# Learning Discriminative and Generalizable Anomaly Detector for Dynamic Graph with Limited Supervision

Yuxing Tian[* 1]  Yiyan Qi[* 2]  Fengran Mo[1]  Weixu Zhang[3 4]  Jian Guo[2]  Jian-Yun Nie[1]

## Abstract

Dynamic graph anomaly detection is critical for many real-world applications but remains challenging due to the scarcity of labeled anomalies. Existing methods are either unsupervised or semi-supervised: unsupervised methods avoid the need for labeled anomalies but often produce ambiguous boundary, whereas semi-supervised methods can overfit to the limited labeled anomalies and generalize poorly to unseen anomalies. To address this gap, we consider a largely underexplored problem: learning a discriminative boundary from normal/unlabeled data, while leveraging limited labeled anomalies **when available** without sacrificing generalization to unseen anomalies. In this paper, we propose an effective, generalizable, and model-agnostic framework with three main components: (i) residual representation encoding that capture deviations between current interactions and their historical context, providing anomaly-relevant signals; (ii) a restriction loss that constrain the normal representations within an interval bounded by two co-centered hyperspheres, ensuring consistent scales while keeping anomalies separable; (iii) a bi-boundary optimization strategy that learns a discriminative and robust boundary using the log-likelihood distribution modeled by a normalizing flow. Extensive experiments demonstrate the superiority of our framework across diverse evaluation settings.

## 1. Introduction

Dynamic graph anomaly detection (DGAD) is vital for real-world applications such as financial fraud detection (Zhang

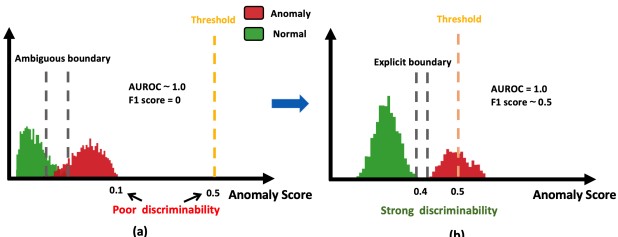

*Figure 1.* Conceptual illustration. (a) Unsupervised methods often yield ambiguous decision boundaries, with anomaly scores collapse into a narrow range. (b) The objective of our framework.

et al., 2021; Li et al., 2023), abnormal social interactions (Berger-Wolf & Saia, 2006; Greene et al., 2010), cyberattacks (Zhang et al., 2022). It has attracted increasing research attention. Previous DGAD methods (Yu et al., 2018; Zheng et al., 2019; Cai et al., 2021; Liu et al., 2023) adopt discrete-time modeling, representing graph evolution as a sequence of snapshots sampled at fixed intervals. This discretization inevitably discards fine-grained temporal information, introducing discretization errors and detection latency. Recent studies (Tian et al., 2023; Postuvan et al., 2024; Yang et al., 2025) instead leverage continuous-time modeling to mitigate these issues. Nevertheless, both settings still face a fundamental challenge: labeled anomalies are far rarer than normal instances in real-world scenarios.

To avoid the need for labeled anomalies, most DGAD methods (Yu et al., 2018; Cai et al., 2021; Liu et al., 2023; Postuvan et al., 2024; Yang et al., 2025) follow an unsupervised paradigm, learning representations from normal or unlabeled samples and identifying anomalies as deviations from learned normal patterns. However, without explicit supervision, the induced decision boundary is often ambiguous and weakly discriminative. As illustrated in Figure 1(a), unsupervised methods typically produce scores that collapse into a narrow low-valued range with substantial overlap, which is also observed in our experiments (refer to Fig. 4). Consequently, the operating threshold becomes inherently under-determined and sensitive to minor distributional changes, making the scores unreliable for consistent binary decisions. This drawback is particularly critical in high-stakes applications such as financial fraud detection,

*Equal contribution  [1]Université de Montréal [2]International Digital Economy Academy (IDEA) [3]McGill University [4]Mila - Quebec Artificial Intelligence Institute. Correspondence to: Jian-Yun Nie <nie@iro.umontreal.ca>.

*Proceedings of the 43rd International Conference on Machine Learning*, Seoul, South Korea. PMLR 306, 2026. Copyright 2026 by the author(s).

where deployment requires stable and well-calibrated decision rules (e.g., fixed or tiered cutoffs) for transaction-level decisions. In practice, it is often possible to identify a limited number of labeled anomalies, which can be exploited by semi-supervised DGAD methods (Zheng et al., 2019; Tian et al., 2023). A typical approach is to perform pseudo-labeling of unlabeled data based on the limited labeled anomalies. However, pseudo labels are inherently noisy, and the limited labeled anomalies can cover only a narrow subset of anomaly patterns. Consequently, the learned decision boundary is skewed toward the observed anomaly patterns, leading to poor generalization to unseen anomalies in open-set scenarios.

These limitations raise a natural question: *How can we develop a DGAD method that can learn an explicit, discriminative decision boundary from normal/unlabeled samples, while maximizing the utility of limited labeled anomalies **when available** without sacrificing generalization to unseen anomalies?* Achieving this objective requires two essential capabilities: First, it should be capable of learning informative representations that provide anomaly-relevant signals with sufficiently discriminability to distinguish anomalous behaviors. Second, it should learn and optimize an explicit boundary that tightly encloses normal representations, pushes anomalies outside, and avoids being biased by the few labeled anomalies.

In this paper, we propose **SDGAD**, an effective and generalizable DGAD framework that can be seamlessly integrated with existing models based on continuous-time dynamic graph (CTDG). Specifically, to learn informative representations, we introduce **residual representation encoding** to explicitly preserve the distinction between the current interaction and its recent context, which is often weakened by history-dominated aggregation in standard CTDG encoders. Concretely, we compute embeddings with and without the current event and take their discrepancy as the residual, which highlights deviations from recent graph evolution and provides an explicit anomaly-relevant signal. However, different anomaly patterns can induce residuals at vastly different scales, which hinders learning a unified boundary. We therefore introduce a **representation restriction** loss that constrains normal residuals to a bounded region defined by two co-centered hyperspheres, while pushing anomalies outside. Finally, we employ a normalizing flow as a density estimator to model the log-likelihood distribution of normal samples, yielding an explicit scoring function for boundary construction. Building on this, we propose a **bi-boundary optimization** strategy to learn a discriminative and robust decision boundary. Our contributions are as follows:

(1) We propose **SDGAD**, an effective and generalizable model-agnostic DGAD framework that performs well under both unsupervised and limited-supervision settings.

(2) We introduce a residual representation encoding mechanism with restriction to learn discriminative representations and a bi-boundary optimization strategy to construct explicit and robust boundaries.

(3) We conduct comprehensive experiments on six datasets with both real and synthetic anomalies under three settings (unsupervised, limited supervision and few-shot). The results demonstrate that our framework consistently outperforms baselines. The code is publicly available at here.

## 2. Related Work

### 2.1. Anomaly Detection in Dynamic Graphs

Dynamic graph anomaly detection (DGAD) methods primarily differ in their temporal modeling assumptions and can be broadly classified as discrete-time (DTDG) or continuous-time (DTDG). DTDG approaches (Yu et al., 2018; Zheng et al., 2019; Cai et al., 2021) discretize graph evolution into snapshots, but fixed-interval sampling often obscures interaction-level dynamics and loses fine-grained temporal signals. CTDG models (Postuvan et al., 2024; Reha et al., 2023; Tian et al., 2023; Yang et al., 2025) instead represent evolution as a timestamped event stream. Nevertheless, it does not resolve the core challenges of dynamic graph anomaly detection. In particular, most existing methods, whether DTDG-based or CTDG-based, adopt unsupervised learning and are trained primarily on normal or unlabeled data. Without explicit supervision to contrast normal and anomalous patterns, they often induce ambiguous and weakly discriminative decision boundaries. In practice, there may be a limited number of labeled anomalies available. Semi-supervised DGAD methods (Zheng et al., 2019; Liu et al., 2023; Tian et al., 2023) incorporate the few available anomaly labels and expand supervision via pseudo-labeling. For instance, SAD (Tian et al., 2023) generates pseudo-labels using a deviation network and optimizes a contrastive objective based on these labels. However, performance can degrade markedly under extreme label scarcity, because the few labeled anomalies often provide only partial and biased coverage of anomaly patterns, leading to overfitting and poor generalization to unseen anomalies. Moreover, noise in pseudo-labeling weakens effective supervision and can bias boundary learning.

### 2.2. Learning under Limited Supervision

The extreme rarity of anomalies in real-world applications results in severe class imbalance and limited positive supervision, posing a fundamental challenge for DGAD. Standard imbalance-learning techniques such as re-sampling and re-weighting (Wang et al., 2019; Cui et al., 2020; Dou et al., 2020; Liu et al., 2020) are often effective for static graphs, but they rely on independent and identically distributed

(i.i.d.) assumptions that are are not generally satisfied in dynamic graphs due to temporal dependencies and evolving interactions. Data augmentation provides an alternative (Hou et al., 2022; Kong et al., 2022; Chen et al., 2024; Zhao et al., 2021), yet most existing augmentations are designed for static graphs and typically depend on node or edge attributes that are sparse or unavailable in dynamic settings. Although recent dynamic-graph augmentations have been proposed (Tian et al., 2024a; Wang et al., 2021b), they largely rely on empirical heuristics and offer limited guarantees that synthesized anomalies match real anomalous behaviors. Consequently, effective learning under severe class imbalance and limited supervision in open-set settings remains an open challenge for dynamic graph anomaly detection (Ma et al., 2023; Qiao et al., 2025; Hua et al., 2026).

## 3. Preliminaries

**Notations.** A continuous-time dynamic graph (CTDG) is formulated as an ordered stream of events $G = \{\xi(t_0) \ldots \xi(t_k) \ldots \xi(t_n)\}_{k=0}^n$ with nondecreasing timestamps $t_0 \leq t_k \leq t_n$. Each event $\xi(t_k) = (v_i, v_j, t_k, e_{i,j}^{t_k})$ represents an interaction from the source node $v_i$ to the destination node $v_j$ at timestamp $t_k$, accompanied by an edge feature $e_{i,j}^{t_k}$. For non-attributed graphs, we set both node and edge features to zero vectors. Multiple interactions can occur either between the same node pair at different timestamps or among different node pairs at the same timestamp. Each interaction can be further associated with a binary label $y \in 0, 1$, where $0$ denotes normal and $1$ denotes anomalous.

**CTDG Encoder.** For DGAD, dynamic node representations constitute the primary features used for anomaly scoring and detection. The CTDG encoder is therefore a core component, as it determines the quality of these representations. Since our framework is expected to work with arbitrary CTDG encoders, we first present a generic formulation of the encoding interface. Formally, given an interaction $\xi(t) = (v_i, v_j, t, e_{i,j}^t)$ [1], let $\mathcal{N}_{<t}(v_i) = \{(v_{n_\ell}, e_{i,n_\ell}^{t_\ell}, t_\ell)\}_{\ell=1}^L$ be the $L$ historical interactions of node $v_i$ sampled from its past, with $t_\ell < t$. We can construct a temporally ordered input sequence $\mathcal{S}_i^t$ as:

$$\mathcal{S}_i^t = \left[ (v_{n_1}, e_{i,n_1}^{t_1}, t_1), \ldots, (v_{n_L}, e_{i,n_L}^{t_L}, t_L), (v_j, e_{i,j}^t, t) \right] \tag{1}$$

Then the CTDG encoder $\text{Enc}(\cdot)$ outputs the time-dependent representation $\mathbf{E}_i^t$ of node $v_i$ at timestamp $t$:

$$\mathbf{E}_i^t = \text{Enc}(\mathcal{S}_i^t) \tag{2}$$

The representation of node $v_j$ is computed analogously. Although the encoding details vary across CTDG models,

---

[1]For notational clarity, we omit the subscript $k$ in all subsequent formulations

this abstraction captures the general procedure. Model-specific implementations are provided in Appendix C.3.

**Problem Definition.** We study anomaly detection on CTDG, where interactions arrive sequentially as an event stream. In realistic settings, anomalies are extremely rare and supervision is limited. Our goal is to develop a framework that can be trained fully unsupervised when labels are unavailable, while also being capable of effectively leveraging a small number of labeled anomalies when available. For each arriving event $\xi(t)$, the framework learns its representation conditioned on the evolving historical graph and outputs a continuous anomaly score $s(\xi(t)) \in [0, 1]$.

## 4. Methodology

Our framework is illustrated in Fig. 2 and consists of three key components: residual representation encoding, representation restriction and bi-boundary optimization. We describe each component in detail in the following subsections.

### 4.1. Residual Representation Encoding

In dynamic graphs, anomalies are often characterized by interactions whose behavior deviates from the recent evolution of the graph. As shown in Eq. 1 and Eq. 2, standard CTDG encoders compute node representations by aggregating historical interactions together with the current event, resulting in embeddings that are largely dominated by long-term context. While this design effectively captures stable temporal dependencies, it can weaken the distinction between the newly observed interaction and its recent context, which is crucial for identifying rare anomalous events. To explicitly preserve this distinction, we introduce residual representation encoding. Specifically, we define the residual representation of node $v_i$ as:

$$\Delta \mathbf{E}_i^t = \mathbf{E}_i^t - \mathbf{E}_i^{t^-} = \text{Enc}(\mathcal{S}_i^t) - \text{Enc}(\mathcal{S}_i^{t-}) \tag{3}$$

where $\mathcal{S}_i^{t-} = \left[ (v_{n_1}, e_{i,n_1}^{t_1}, t_1), \ldots, (v_{n_L}, e_{i,n_L}^{t_L}, t_L) \right]$ denotes the historical interaction sequence of $v_i$ before time $t$. The residual representation $\Delta \mathbf{E}_i^t$ captures the incremental change in the node representation introduced by the current interaction. When the new interaction is consistent with the historical interaction patterns of $v_i$, this change is expected to be small; in contrast, anomalous interactions tend to induce a larger deviation. This design is consistent with the neighborhood-consistency principle in graph analysis, where normal behavior exhibits local continuity, whereas anomalies break such continuity and lead to pronounced representation shifts. By subtracting the stable historical component and highlighting the event-driven variation, the residual representation provides a more discriminative signal for anomaly detection. Finally, we construct

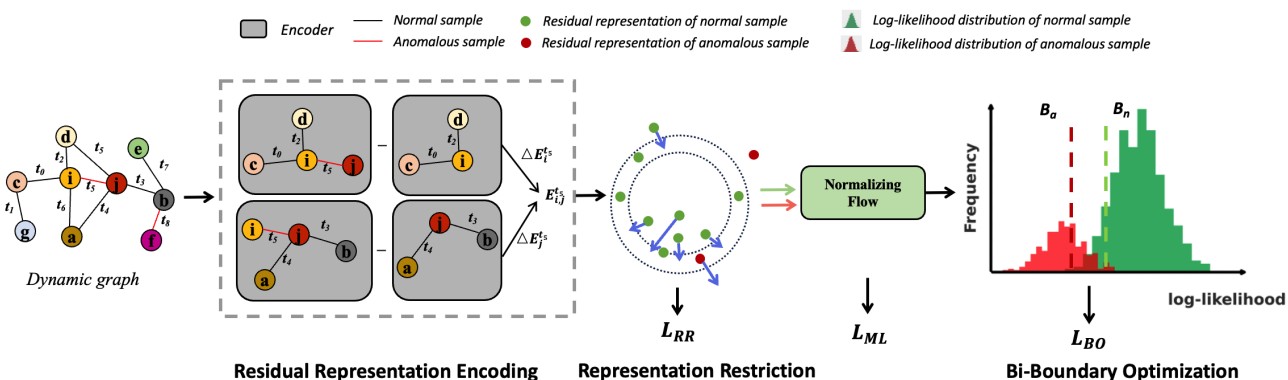

*Figure 2.* Framework overview. We first encode each sample's residual representation by contrasting two node-pair embeddings (Sec 4.1). Then the residual representations of normal samples (green dots) are constrained into an interval region between two co-centered hypersphere, while anomalous samples (red dots) are pushed outside (Sec 4.2). Finally, normalizing flow is used to model the normal log-likelihood distribution and then a discriminative and robust boundary is explicitly learned by the bi-boundary optimization(Sec 4.3).

the event-level representation by concatenating the residual representations of the two incident nodes:

$$\mathbf{E}_{i,j}^t = \Delta\mathbf{E}_i^t \| \Delta\mathbf{E}_j^t. \tag{4}$$

## 4.2. Representation Restriction

Although residual representations can provide informative anomaly-relevant signals, their effectiveness is constrained by the diversity of anomaly patterns induced by different temporal dynamics and structural contexts: some anomalies trigger pronounced deviations (e.g., interaction bursts), while others manifest as subtle shifts (e.g., mild timing irregularities). This variability makes it challenging to learn a unified decision boundary that generalizes across patterns.

Thus, we introduce a representation restriction loss that increases the separation between normal and anomalous representations. The design is motivated by one-class classification (Ruff et al., 2018), which constrains normal representations to lie inside an origin-centered hypersphere. However, Zhang et al. (2024) show that this assumption can be fragile: normal representations often concentrate near the boundary of the hypersphere rather than spread across its interior, leaving the inner region weakly supported. This effect is known as the soap-bubble phenomenon (Vershynin, 2018). Consequently, the learned normal region may become sparse and permissive, allowing anomalous samples to enter the same region and thereby reducing discriminability.

Therefore, Zhang et al. (2024) propose using two co-centered hyperspheres to provide tighter support. However, it is built on the assumption that samples are independent and identically distributed (i.i.d.) and requires the evaluation data to satisfy the same i.i.d. assumption. In contrast, samples in dynamic graphs are generally non-i.i.d., which makes the hypersphere learning strategy of Zhang et al. (2024) unsuitable for our setting. Also, its bi-hypersphere

construction is computationally expensive, is not end-to-end. Therefore, constructing an effective bi-hypersphere in an end-to-end manner for dynamic graph anomaly detection is a new technical challenge.

Specifically, we first project the residual representation $E_{i,j}^t$ through a linear layer to obtain projected representation $x$. To robustly measure its distance to the origin, we adopt the pseudo-Huber norm surrogate:

$$n(x) = \sqrt{\|x\|_2^2 + 1} - 1 \tag{5}$$

which smoothly transitions from quadratic to linear penalization. We then construct the two co-centered hyperspheres with an outer radius $r_{\max}$ and an inner radius $r_{\min} = \gamma \cdot r_{\max}$, where $\gamma$ is a coefficient (e.g. 0.9), to form a compact normal region. In practice, we set $r_{\max} = 0.4$ since $n(x) \approx 0.4$ when $\|x\|_2^2 = 1$, which is consistent with the unit hypersphere under the Euclidean norm. Then, we can define the following loss for normal samples:

$$\mathcal{L}_n(x) = \begin{cases} -\sigma(-(\Delta_{min})) \cdot e^{\Delta_{min}}, & n(x) \le r_{min} \\ -\sigma(-(\Delta_{max})) \cdot e^{\Delta_{max}}, & n(x) \ge r_{max} \\ 0, & otherwise \end{cases} \tag{6}$$

where $\sigma(\cdot)$ denotes the log-sigmoid function, $\Delta_{\min} = r_{\min} - n(x)$, and $\Delta_{\max} = n(x) - r_{\max}$. The objective of this loss is to pull the representations of normal samples into a compact interval region at a consistent scale, while avoiding unnecessary contraction for samples already inside, thereby preventing representation collapse and enabling a unified decision boundary.

For abnormal representations, we expect that they are located outside the outer hypersphere and are highly different from normal representations. Thus, we contrastively optimize the norms of abnormal representations to form a

gap between abnormal and normal representations. We introduce a margin $\Delta r$ and define an abnormal radius $r' = r_{max} + \Delta r$. When an abnormal representation $x$ is inside the hypersphere with a radius $r'$, it will be pushed outside of the hypersphere. In addition, to avoid overfitting to anomalies during pretraining, we do not further push away the abnormal representations that are already outside the hypersphere. Then the learning objective is as follows:

$$\mathcal{L}_a(x) = \begin{cases} -\sigma(-(r' - n(x))) \cdot e^{r' - n(x)}, & n(x) \leq r' \\ 0, & n(x) > r' \end{cases}$$
(7)

This loss enforces separation by penalizing anomalies that fall within the interval region. Thus, combining the two learning objectives, we call the final loss as representation restriction loss, which is defined as:

$$\mathcal{L}_{\mathcal{RR}} = \mathbb{I}_{[y=0]} \cdot \mathcal{L}_n(x) + \mathbb{I}_{[y=1]} \cdot \mathcal{L}_a(x)$$
(8)

In Eq.(8), $x$ can be either normal or anomaly.

### 4.3. Bi-Boundary Optimization

Building on the restricted representations, we then employ nomalizing flow as the distribution estimator to model the normal distribution and treat anomalies as out-of-distribution deviations. Notably, our framework is not tied to normalizing flows, other generative models can be used in its place. But for this task, a desirable density estimator would (i) provide explicit per-sample likelihoods for anomaly scoring, (ii) remain tractable during both training and inference, and (iii) be sufficiently expressive to model complex non-Gaussian feature distributions. Under these criteria, normalizing flows provide a practical balance among tractability, modeling capacity, and explicit likelihood estimation.

Specifically, normalizing flow provides exact density estimation by transforming an intractable data distribution $\mathcal{Q}$ to a tractable latent distribution $\mathcal{Z}$ via an invertible mapping. It is implemented as a composition of $F$ invertible transformations: $\boldsymbol{\Phi}_\theta = \boldsymbol{\Phi}_F \circ \cdots \circ \boldsymbol{\Phi}_1$, where $\theta$ denotes the trainable parameters. For an input $x$, its log-likelihood $\log[p(x)]$ can be computed as follow:

$$\log[p(x)] = \log p_{\mathcal{Z}}(\boldsymbol{\Phi}_\theta(x)) + \sum_{f=1}^{F} \log \left| \det J_{\boldsymbol{\Phi}_f}(x_{f-1}) \right|$$
(9)

where $J_{\boldsymbol{\Phi}_f}(x_{f-1}) = \frac{\partial \boldsymbol{\Phi}_f(x_{f-1})}{\partial x_{f-1}}$ is the Jacobian matrix, $\det$ denotes the determinant. The latent distribution $\mathcal{Z}$ is typically chosen as a standard Gaussian, and $\theta$ is optimized by maximizing the log-likelihood over the distribution $\mathcal{Q}$. Thus

the resulting maximum-likelihood objective is given by:

$$\mathcal{L}_{\mathcal{ML}} = \mathbb{E}_{x \sim \mathcal{Q}} \Big[ \tfrac{d}{2} \log(2\pi) + \tfrac{1}{2} \boldsymbol{\Phi}_\theta(x)^T \boldsymbol{\Phi}_\theta(x) \\ - \sum_{f=1}^{F} \log \left| \det J_{\boldsymbol{\Phi}_f}(x_{f-1}) \right| \Big]$$
(10)

Since log-likelihood values can vary widely and are unbounded below, we rescale them into $[-1, 0]$ with a normalization constant for more stable optimization.

As training is conducted batch-wise, we denote the log-likelihood of normal and anomalous samples within a batch as $\mathcal{D}_n = \{\log[p(x_i)]\}_{i=1}^N$ and $\mathcal{D}_a = \{\log[p(x_j)]\}_{j=1}^M$, where $N$ and $M$ denote their respective counts (Note that as labeled anomalies are limited, many batches have $M = 0$). We treat $\mathcal{D}_n$ as a batch-level estimate of the normal log-likelihood distribution and define a decision boundary $\mathcal{B}$ accordingly, which serves as the reference for anomaly identification. A straightforward way is to set $\mathcal{B}$ as the $\alpha$-th percentile of sorted normal log-likelihood distribution $\mathcal{D}_n$ and identify samples with lower values as anomalies. However, in high-dimensional spaces, probability mass often concentrates around the *typical set* rather than in the low-density tail (Kirichenko et al., 2020), and the invertible nature of normalizing flows can consequently assign high likelihood to out-of-distribution inputs by mapping them into typical latent regions (Kumar et al., 2021). As a result, anomalous samples may not exhibit low log-likelihoods and can even attain higher values than normal samples.

To mitigate this issue, we propose a bi-boundary optimization strategy. Specifically, we introduce a decision margin $\tau$ (e.g., $\tau = 0.1$) and split the single boundary $\mathcal{B}$ into a normal boundary $\mathcal{B}_n$ and an anomalous boundary $\mathcal{B}_a = \mathcal{B}_n - \tau$. The margin between $\mathcal{B}_n$ and $\mathcal{B}_a$ defines a buffer region that reduces boundary ambiguity and improves robustness by enforcing separation between normal and anomalous regions. The resulting objective is:

$$\mathcal{L}_{\mathcal{BO}} = \sum_{i=1}^{N} |\min(softplus(\log[p(x_i)] - \mathcal{B}_n), 0)| + \\ \sum_{j=1}^{M} |\max(softplus(\log[p(x_j)] - \mathcal{B}_a), 0)|$$
(11)

Unlike hard losses that introduce discontinuous, non-differentiable penalties at the boundary, we adopt the softplus function, $softplus(\cdot) = \log(1 + e^x)$, which yields smooth and violation-aware penalties. Minimizing $\mathcal{L}_{\mathcal{BO}}$ promotes a clear margin between normal and anomalous log-likelihoods by constraining anomalous samples to $(-\infty, \mathcal{B}_a]$ and normal samples to $[\mathcal{B}_n, 0]$. Samples that violate these constraints are penalized proportionally to the magnitude of their violations, encouraging normal samples

to concentrate in high-density regions while pushing anomalies further away.

## 4.4. Overall Training Objective

The overall training loss of our framework is the combination of the Eq. 10, Eq. 8 and Eq. 11 as follows:

$$\mathcal{L} = \mathcal{L}_{\mathcal{ML}} + \lambda_1 \mathcal{L}_{\mathcal{BO}} + \lambda_2 \mathcal{L}_{\mathcal{RR}} \qquad (12)$$

Here $\mathcal{L}_{ML}$ denotes the maximum-likelihood objective of the normalizing flow, computed only on normal samples to model the normal distribution. The coefficients $\lambda_1$ and $\lambda_2$ weight the contributions of $\mathcal{L}_{BO}$ and $\mathcal{L}_{RR}$, respectively. We further study the sensitivity of balancing among the three loss terms in Appendix E.1, and provide a corresponding error bound analysis in Appendix E.2.

## 4.5. Anomaly Scoring

We define the anomaly score for each sample as the complement of its log-likelihood, where higher values correspond to stronger deviations from the normal distribution. Owing to the monotonicity of the exponential function $exp(\cdot)$, the anomaly score for a sample $x$ can be written as:

$$s(x) = 1 - \exp(\log[p(x)]) \qquad (13)$$

## 5. Experiments

### 5.1. Experimental Setup

**Datasets.** We evaluate on six datasets, including three with real-world labeled anomalies (*Wikipedia*, *Reddit*, and *MOOC*) and three benign datasets with injected anomalies (*Enron*, *UCI*, and *LastFM*). Detailed descriptions, statistics, and preprocessing procedures are provided in Appendix B.

**Baselines.** We compare our framework against 15 baselines, grouped into three categories: (1) DTDG-based methods designed for DGAD, including Netwalk (Yu et al., 2018), AddGraph (Zheng et al., 2019), StrGNN (Cai et al., 2021) and TADDY (Liu et al., 2023). (2) CTDG-based methods designed for DGAD, including SAD (Tian et al., 2023) and GeneralDyG (Yang et al., 2025) (3) Representative CTDG models, including JODIE (Kumar et al., 2019), DyRep (Trivedi et al., 2019), TGAT (Xu et al., 2020), TGN (Rossi et al., 2020), TCL (Wang et al., 2021a), GraphMixer (Cong et al., 2023), CAWN (Wang et al., 2021c), DyGFormer (Yu et al., 2023), and FreeDyG (Tian et al., 2024b). We provide descriptions and implementation details in Appendix C.

**Evaluation Metrics.** Most previous studies rely solely on Area Under the Receiver Operating Characteristic Curve (AUROC) as the metric. However, as discussed in Davis & Goadrich (2006) and further analyzed in Appendix F.1, AUROC is overly optimistic under highly imbalanced label distributions. We therefore also report two additional metrics for a more comprehensive evaluation: Area Under the Prevision Recall Curve (AP) and F1 score.

**Settings.** To provide a rigorous and comprehensive comparison, we consider three evaluation settings: **S1 (limited supervision):** all available anomaly labels in the training set can be used for model training. **S2 (few-shot):** only ($k \in \{1, 2, 3\}$) labeled anomalies, randomly sampled from the training set, can be used for model training. **S3 (fully unsupervised):** models are trained without any label supervision. For **S1** and **S2**, only semi-supervised methods (TADDY, AddGraph, and SAD) natively incorporates label supervision. To make the supervision setting comparable across methods, we follow (Tian et al., 2023) and attach the same MLP classifier to all baselines that lack built-in supervision, training it with a cross-entropy objective.

### 5.2. Experimental Analysis

**Main Results.** Table 1 reports results for all baselines on datasets with real-world anomalies under the **S1** setting. DTDG-based methods perform the worst across all datasets, since snapshot-level modeling inevitably discards fine-grained temporal dependencies that are crucial for capturing anomalous behaviors in dynamic graphs. In contrast, CTDG-based DGAD methods and representative CTDG encoders achieve substantially stronger performance, with only a small gap between them. This suggests that existing task-specific DGAD designs provide limited gains beyond general-purpose CTDG representations, likely because both mainly focus on temporal-structural representation learning rather than learning robust decision boundaries for anomaly detection. Overall, our framework consistently outperforms all baselines. Using TCL as the CTDG backbone, it delivers a substantial F1 improvement while maintaining strong AUROC and AP. Similar gains with CAWN and DyGFormer further indicate that the improvements are robust and not tied to a specific encoder design.

We also conduct experiments on datasets with synthetic anomalies under the **S1** and **S3** settings, with results reported in Table. 11 and Table. 2, respectively. Notably, synthetic anomalies can include both normal and abnormal interactions between the same node pair within a single snapshot. Because DTDG methods discard the temporal order of interactions within each snapshot, they cannot resolve these within-snapshot conflicts and are therefore unsuitable for this setting. Remarkably, in the fully unsupervised **S3** setting, our framework achieves performance comparable to that in the limited-supervision **S1** setting. Although most baselines benefit from limited supervision, their gains often

| Methods | Wikipedia | | | Reddit | | | MOOC | | |
|---|---|---|---|---|---|---|---|---|---|
| | AUROC | AP | F1 | AUROC | AP | F1 | AUROC | AP | F1 |
| DTDG-based DGAD Methods | | | | | | | | | |
| Netwalk | $73.10_{\pm2.12}$ | $1.28_{\pm1.14}$ | $0_{\pm0}$ | $59.18_{\pm2.02}$ | $0.09_{\pm0.04}$ | $0_{\pm0}$ | $64.12_{\pm0.98}$ | $2.21_{\pm0.43}$ | $0_{\pm0}$ |
| AddGraph | $74.80_{\pm1.98}$ | $1.63_{\pm1.19}$ | $0.92_{\pm0.15}$ | $58.37_{\pm4.28}$ | $0.12_{\pm0.05}$ | $0_{\pm0}$ | $66.35_{\pm1.76}$ | $2.52_{\pm0.39}$ | $0_{\pm0}$ |
| STRGNN | $72.87_{\pm3.31}$ | $2.24_{\pm2.01}$ | $0_{\pm0}$ | $59.26_{\pm3.14}$ | $0.10_{\pm0.03}$ | $0_{\pm0}$ | $63.47_{\pm2.05}$ | $1.98_{\pm0.37}$ | $0_{\pm0}$ |
| Taddy | $75.40_{\pm2.88}$ | $2.52_{\pm1.41}$ | $1.34_{\pm0.17}$ | $61.04_{\pm2.33}$ | $0.14_{\pm0.06}$ | $0.10_{\pm0.05}$ | $67.02_{\pm1.64}$ | $3.83_{\pm0.41}$ | $0_{\pm0}$ |
| CTDG-based DGAD Methods | | | | | | | | | |
| SAD | $79.84_{\pm1.91}$ | $5.12_{\pm2.03}$ | $4.25_{\pm3.84}$ | $62.98_{\pm2.05}$ | $0.17_{\pm0.05}$ | $0.76_{\pm0.48}$ | $71.25_{\pm0.77}$ | $6.74_{\pm0.52}$ | $3.12_{\pm1.21}$ |
| GeneralDyG | $77.52_{\pm1.05}$ | $3.15_{\pm0.87}$ | $1.06_{\pm1.41}$ | $61.43_{\pm1.48}$ | $0.15_{\pm0.03}$ | $0_{\pm0}$ | $70.12_{\pm0.83}$ | $5.31_{\pm0.64}$ | $0_{\pm0}$ |
| Representative CTDG Models | | | | | | | | | |
| JODIE | $80.23_{\pm1.39}$ | $1.87_{\pm1.02}$ | $0_{\pm0}$ | $55.93_{\pm5.06}$ | $0.13_{\pm0.03}$ | $0_{\pm0}$ | $72.12_{\pm0.84}$ | $2.40_{\pm0.64}$ | $0_{\pm0}$ |
| DyRep | $83.89_{\pm1.03}$ | $2.60_{\pm0.31}$ | $0_{\pm0}$ | $58.83_{\pm2.95}$ | $0.14_{\pm0.02}$ | $0_{\pm0}$ | $72.21_{\pm0.25}$ | $3.56_{\pm0.04}$ | $0_{\pm0}$ |
| TGN | $85.51_{\pm1.12}$ | $3.80_{\pm1.47}$ | $0.96_{\pm1.32}$ | $64.31_{\pm0.72}$ | $0.14_{\pm0.01}$ | $0_{\pm0}$ | $76.63_{\pm0.98}$ | $6.47_{\pm0.15}$ | $0_{\pm0}$ |
| TGAT | $76.93_{\pm1.14}$ | $2.78_{\pm1.04}$ | $0.46_{\pm1.03}$ | $61.58_{\pm1.72}$ | $0.13_{\pm0.01}$ | $0_{\pm0}$ | $69.05_{\pm0.92}$ | $4.86_{\pm0.41}$ | $0_{\pm0}$ |
| TCL | $77.69_{\pm0.74}$ | $5.38_{\pm1.21}$ | $2.41_{\pm0.0}$ | $60.47_{\pm2.13}$ | $0.22_{\pm0.21}$ | $0_{\pm0}$ | $72.51_{\pm1.76}$ | $7.87_{\pm0.71}$ | $0_{\pm0}$ |
| CAWN | $78.97_{\pm0.56}$ | $4.96_{\pm0.72}$ | $1.44_{\pm1.32}$ | $65.29_{\pm0.92}$ | $0.18_{\pm0.03}$ | $0.14_{\pm0.32}$ | $72.63_{\pm0.39}$ | $7.58_{\pm0.24}$ | $0_{\pm0}$ |
| GraphMixer | $76.19_{\pm2.29}$ | $2.80_{\pm1.65}$ | $1.67_{\pm3.73}$ | $60.11_{\pm3.61}$ | $0.13_{\pm0.02}$ | $0_{\pm0}$ | $71.03_{\pm0.52}$ | $5.32_{\pm1.07}$ | $0_{\pm0}$ |
| DyGFormer | $85.58_{\pm1.25}$ | $2.58_{\pm0.72}$ | $0.48_{\pm1.06}$ | $66.70_{\pm2.09}$ | $0.25_{\pm0.11}$ | $0_{\pm0}$ | $72.63_{\pm0.33}$ | $6.20_{\pm0.36}$ | $0_{\pm0}$ |
| FreeDyG | $77.22_{\pm4.21}$ | $3.01_{\pm1.19}$ | $1.26_{\pm0.73}$ | $63.99_{\pm2.76}$ | $0.19_{\pm0.04}$ | $0_{\pm0}$ | $73.10_{\pm0.71}$ | $5.91_{\pm0.92}$ | $0_{\pm0}$ |
| Ours | | | | | | | | | |
| SDGAD (TCL) | $80.36_{\pm0.69}$ | $\mathbf{5.41}_{\pm1.23}$ | $\mathbf{8.34}_{\pm2.72}$ | $62.22_{\pm0.95}$ | $0.49_{\pm0.26}$ | $2.74_{\pm1.86}$ | $72.87_{\pm1.40}$ | $7.89_{\pm0.38}$ | $\mathbf{7.15}_{\pm0.64}$ |
| SDGAD (CAWN) | $80.84_{\pm0.65}$ | $5.15_{\pm0.79}$ | $7.41_{\pm3.04}$ | $66.81_{\pm1.08}$ | $0.57_{\pm0.05}$ | $\mathbf{3.28}_{\pm2.02}$ | $73.02_{\pm0.41}$ | $\mathbf{8.62}_{\pm0.25}$ | $6.74_{\pm0.53}$ |
| SDGAD (DyGFormer) | $\mathbf{86.60}_{\pm1.20}$ | $3.71_{\pm0.77}$ | $4.15_{\pm2.06}$ | $\mathbf{67.24}_{\pm1.12}$ | $\mathbf{0.88}_{\pm0.11}$ | $3.15_{\pm1.11}$ | $\mathbf{73.25}_{\pm0.36}$ | $6.39_{\pm0.48}$ | $5.86_{\pm0.77}$ |

*Table 1.* Performance comparison ($Avg_{\pm std}$ in %) on datasets with real anomalies under **S1**. Best results are highlighted in **green**.

| Methods | UCI | | | Enron | | | LastFM | | |
|---|---|---|---|---|---|---|---|---|---|
| | AUROC | AP | F1 | AUROC | AP | F1 | AUROC | AP | F1 |
| CTDG-based DGAD Methods | | | | | | | | | |
| SAD | $86.57_{\pm6.70}$ | $16.90_{\pm5.18}$ | $0_{\pm0}$ | $79.59_{\pm3.82}$ | $3.58_{\pm2.97}$ | $0_{\pm0}$ | $85.37_{\pm1.33}$ | $1.79_{\pm0.40}$ | $0_{\pm0}$ |
| GeneralDyG | $82.25_{\pm7.72}$ | $7.63_{\pm8.86}$ | $4.85_{\pm10.84}$ | $84.26_{\pm4.16}$ | $5.90_{\pm3.11}$ | $0_{\pm0}$ | $83.84_{\pm0.24}$ | $8.46_{\pm0.02}$ | $0_{\pm0}$ |
| Representative CTDG Models | | | | | | | | | |
| JODIE | $56.58_{\pm8.30}$ | $0.44_{\pm0.14}$ | $0.15_{\pm0.33}$ | $37.40_{\pm4.64}$ | $1.30_{\pm0.31}$ | $0_{\pm0}$ | $40.83_{\pm4.64}$ | $3.46_{\pm0.74}$ | $0_{\pm0}$ |
| DyRep | $65.60_{\pm7.70}$ | $0.58_{\pm0.14}$ | $0_{\pm0}$ | $57.40_{\pm9.83}$ | $2.55_{\pm0.54}$ | $0_{\pm0}$ | $58.05_{\pm2.34}$ | $4.41_{\pm0.93}$ | $0_{\pm0}$ |
| TGN | $84.33_{\pm5.30}$ | $14.34_{\pm4.80}$ | $0_{\pm0}$ | $74.06_{\pm5.09}$ | $1.21_{\pm1.36}$ | $0_{\pm0}$ | $75.90_{\pm0.53}$ | $4.24_{\pm0.56}$ | $0_{\pm0}$ |
| TGAT | $87.22_{\pm8.50}$ | $25.73_{\pm10.23}$ | $0_{\pm0}$ | $78.24_{\pm8.96}$ | $4.15_{\pm3.61}$ | $0_{\pm0}$ | $84.58_{\pm1.95}$ | $0.89_{\pm0.26}$ | $0_{\pm0}$ |
| TCL | $73.02_{\pm7.62}$ | $4.77_{\pm1.67}$ | $0_{\pm0}$ | $57.85_{\pm3.85}$ | $0.79_{\pm0.14}$ | $0_{\pm0}$ | $70.84_{\pm2.43}$ | $7.57_{\pm0.11}$ | $0_{\pm0}$ |
| CAWN | $79.12_{\pm5.74}$ | $9.57_{\pm2.56}$ | $0_{\pm0}$ | $76.81_{\pm3.85}$ | $3.42_{\pm1.94}$ | $0_{\pm0}$ | $81.75_{\pm0.15}$ | $1.28_{\pm0.44}$ | $0_{\pm0}$ |
| GraphMixer | $85.42_{\pm5.33}$ | $3.83_{\pm5.86}$ | $0_{\pm0}$ | $81.70_{\pm1.06}$ | $0.96_{\pm0.28}$ | $0_{\pm0}$ | $83.43_{\pm0.78}$ | $5.02_{\pm2.23}$ | $0_{\pm0}$ |
| DyGFormer | $71.11_{\pm4.11}$ | $0.48_{\pm0.36}$ | $0_{\pm0}$ | $64.58_{\pm0.70}$ | $0.28_{\pm0.05}$ | $0_{\pm0}$ | $64.81_{\pm5.22}$ | $0.18_{\pm0.02}$ | $0_{\pm0}$ |
| FreeDyG | $79.09_{\pm6.10}$ | $4.50_{\pm2.76}$ | $0_{\pm0}$ | $76.42_{\pm0.85}$ | $1.02_{\pm0.52}$ | $0_{\pm0}$ | $75.86_{\pm0.64}$ | $3.85_{\pm1.56}$ | $0_{\pm0}$ |
| Ours | | | | | | | | | |
| SDGAD (TCL) | $\mathbf{99.88}_{\pm0.01}$ | $28.85_{\pm0.92}$ | $\mathbf{44.44}_{\pm0.01}$ | $99.86_{\pm0.01}$ | $28.73_{\pm0.33}$ | $44.43_{\pm0.00}$ | $\mathbf{99.87}_{\pm0.01}$ | $\mathbf{29.22}_{\pm0.16}$ | $44.73_{\pm0.10}$ |
| SDGAD (DyGFormer) | $99.88_{\pm0.01}$ | $28.87_{\pm0.32}$ | $27.22_{\pm6.24}$ | $99.87_{\pm0.01}$ | $\mathbf{28.83}_{\pm0.45}$ | $39.87_{\pm2.55}$ | $99.87_{\pm0.01}$ | $28.90_{\pm1.24}$ | $\mathbf{44.75}_{\pm0.25}$ |
| SDGAD (CAWN) | $99.88_{\pm0.01}$ | $\mathbf{29.35}_{\pm1.10}$ | $33.53_{\pm9.97}$ | $\mathbf{99.87}_{\pm0.01}$ | $28.57_{\pm0.74}$ | $\mathbf{44.64}_{\pm0.00}$ | $99.87_{\pm0.01}$ | $28.82_{\pm0.32}$ | $44.68_{\pm0.11}$ |

*Table 2.* Performance comparison ($Avg_{\pm std}$ in %) on datasets with synthetic anomalies under **S3** . Best results are highlighted in **green**.

come with larger run-to-run variance, indicating unstable training. In contrast, our framework delivers consistently strong performance with smaller standard deviations. Moreover, replacing the underlying encoder still yields stable gains, confirming that the improvements are robust and not tied to a specific encoder design.

**Few-shot Setting.** We further evaluate our framework under the **S2** (few-shot) setting to assess its effectiveness in a more extreme limited-supervision scenario than **S1**. Due to page limitations, the results are deferred to Appendix H (Table. 12 and Table. 13). We observe that our framework remains consistently effective even when only $k \in \{1, 2, 3\}$

labeled anomalies are available: as more anomaly labels are provided, the performance improves steadily across both datasets and different CTDG backbones. In contrast, although several baselines can benefit from additional labels, their gains are often less consistent and accompanied by noticeably standard deviation, indicating unstable training behavior under such limited supervision.

### 5.3. Ablation Study

To validate the contribution of each component, we construct three variants: (1) **w/o Res**, which removes residual representation encoding (2) **w/o $\mathcal{L}_{\mathcal{RR}}$**, which removes the representation restriction loss and (3) **w/o $\mathcal{L}_{\mathcal{BO}}$**, which re-

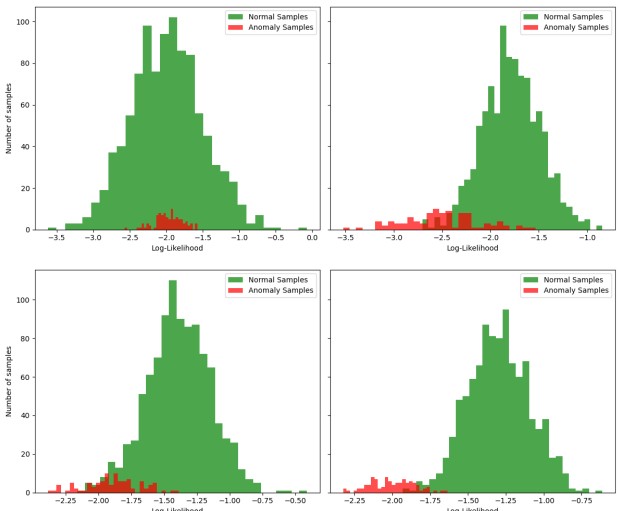

*Figure 3.* Visualization of log-likelihood distributions under different ablation variants. **Top:** baseline without any components (**Left**) vs. adding residual representations without restriction (**Right**). **Bottom:** representation restriction without bi-boundary optimization (**Left**) vs. full framework (**Right**).

| Variant | Wikipedia | | | MOOC | | |
|---|---|---|---|---|---|---|
| | AUROC | AP | F1 | AUROC | AP | F1 |
| SDGAD | **80.36**±0.69 | **5.41**±1.23 | **8.34**±2.72 | **72.87**±1.40 | **7.89**±0.38 | **7.15**±0.64 |
| w/o *Res* | 77.79±3.17 | 3.83±2.48 | 5.29±5.69 | 69.14±2.48 | 5.14±2.48 | 6.58±2.48 |
| w/o $\mathcal{L}_{\mathcal{RR}}$ | 79.44±0.80 | 5.33±1.47 | 6.01±0.72 | 71.02±1.28 | 7.41±0.62 | 5.96±0.81 |
| w/o $\mathcal{L}_{\mathcal{BO}}$ | 80.36±1.00 | 5.38±1.32 | 7.41±3.29 | 72.54±1.35 | 7.56±0.55 | 6.42±0.97 |
| TCL | 77.69±0.74 | 5.28±1.21 | 2.41±0.00 | 72.51±1.76 | 7.87±0.71 | 0±0 |

*Table 3.* Ablation studies on Wikipedia and MOOC. Highlighted are the results ranked first, second, and third.

places the bi-boundary optimization with single-boundary optimization. As shown in Table 3, **w/o *Res*** leads to the most severe degradation, with consistent drops across metrics and larger standard deviations, confirming the importance of residual representations. For **w/o $\mathcal{L}_{\mathcal{RR}}$**, AUROC and AP remain similar, but F1 drops noticeably, since removing the restriction makes normal residuals scale-inconsistent and blurs the boundary for small-residual anomalies. For **w/o $\mathcal{L}_{\mathcal{BO}}$**, AUROC and AP change marginally, while F1 decreases and becomes less stable, indicating increased ambiguity near the decision boundary. Bi-boundary optimization mitigates this by enforcing an explicit margin for more robust thresholding.

## 5.4. Visualization Analysis

To provide an intuitive view of the effect of each component, we visualize the log-likelihood distributions of normal and anomalous samples under different variants in Fig. 3. Experiments are conducted on *Wikipedia* with TCL as the base encoder. In the **top-left** subfigure, the baseline shows

substantial overlap between normal and anomalous distributions, indicating weak discriminability. In the **top-right** subfigure, adding residual representations without the restriction loss (**w/o $\mathcal{L}_{\mathcal{RR}}$**) reduces the overlap, but the normal distribution remains dispersed. The **bottom-left** subfigure corresponds to **w/o $\mathcal{L}_{\mathcal{BO}}$**: the normal distribution becomes more compact and separation improves, yet some anomalies still concentrate near the boundary due to the lack of an explicit likelihood-space margin. Finally, in the **bottom-right** subfigure, the full framework yields a sharply concentrated normal distribution at high likelihood, while anomalies shift toward low-likelihood regions, forming a clear boundary.

We further provide a comparison in Fig. 4 and Fig. 5 to illustrate how anomaly scores differ between TCL and our framework. On the Wikipedia test set, TCL (Fig. 4(a)(c)) produces scores heavily compressed near zero, making threshold selection unstable and sensitive to small perturbations. The corresponding KDE curves also show substantial overlap, suggesting limited separability. In contrast, our framework (Fig. 4(b)(d)) produces well-separated score distributions: normal samples concentrate at low scores, anomalies shift to higher scores, and the KDE curves show minimal overlap. This indicates a tighter characterization of normal behavior and a stronger response to deviations, yielding a more informative and stable score distribution for reliable ranking and threshold-based anomaly detection.

## 5.5. Hyperparameter Study

We analyze four key hyper-parameters in this section. The first is the decision margin $\tau$ used in bi-boundary optimization. The second is the coefficient $\gamma$ used in the representation restriction loss, which determines the strength of the restriction. Third is $\alpha$ sets the normal boundary as the $\alpha$-th percentile of the sorted normal log-likelihood distribution $\mathcal{D}_n$ And $L$ controls the number of sampled historical neighbors during the residual representation encoding stage. Specifically, we conduct experiments with TCL as the encoder on datasets with real anomalies under the **S1** setting and on datasets with synthetic anomalies under the **S3** setting. The results on datasets with real anomalies are reported in Tables 4 and Table 14, respectively.

The results in Table 4a show that the decision margin $\tau$ has a non-trivial impact on performance. In particular, $\tau = 0.1$ yields the strongest results across metrics. A too-narrow margin ($\tau = 0.05$) leaves insufficient room for separation and reduces F1, while an overly wide margin ($\tau = 0.2$) relaxes the anomaly boundary too much and misclassifies borderline normal samples as anomalies. A similar trend holds for $\gamma$ in Table 4b. Larger $\gamma$ enforces stronger consistency among normal samples and generally improves separability, whereas smaller $\gamma$ increases flexibility but weakens separation from anomalies, indicating that $\gamma$ is important

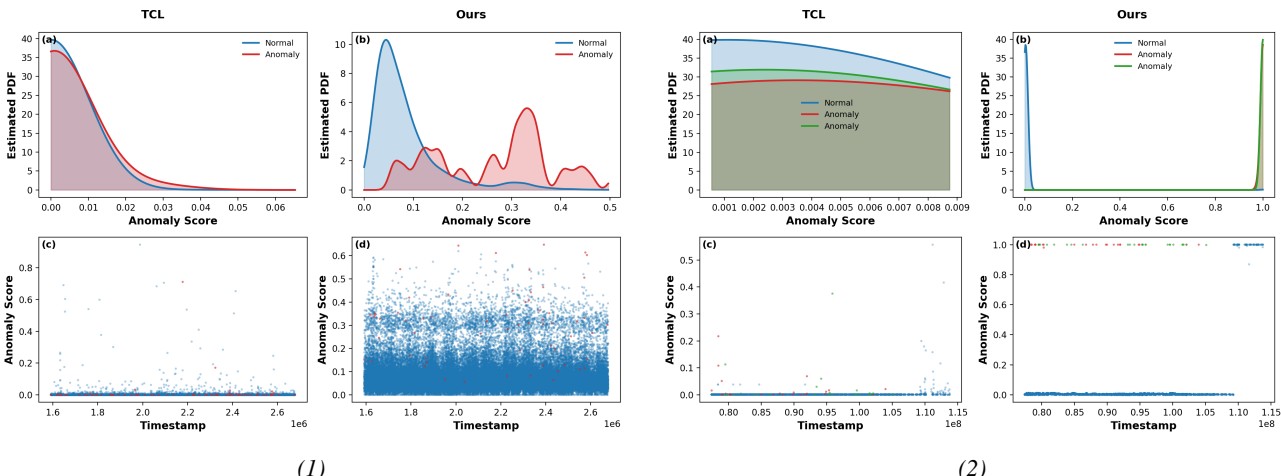

*Figure 4.* Visualization of anomaly score distributions on the Wikipedia (1) and Enron (2) test set. (a–b) Kernel Density Estimates (KDE) of anomaly scores. (c–d) Corresponding anomaly score of each sample over time. Our framework provides significant better score separability and threshold-ability. Results on other datasets are provided in Fig. 5 in the Appendix.

| $\tau$ | Wikipedia | | | MOOC | | |
|---|---|---|---|---|---|---|
| | AUROC | AP | F1 | AUROC | AP | F1 |
| 0.05 | $80.26\pm0.95$ | $4.52\pm1.21$ | $5.06\pm3.49$ | $71.92\pm1.10$ | $7.10\pm0.18$ | $5.77\pm0.85$ |
| 0.1 | $\underline{80.36\pm0.69}$ | $\underline{5.41\pm1.23}$ | $\underline{8.34\pm2.72}$ | $\underline{72.87\pm1.40}$ | $\underline{7.89\pm0.38}$ | $\underline{7.15\pm0.64}$ |
| 0.15 | $80.18\pm1.15$ | $5.08\pm1.60$ | $5.72\pm2.62$ | $72.32\pm1.22$ | $6.67\pm1.13$ | $5.77\pm1.44$ |
| 0.2 | $77.26\pm6.33$ | $3.57\pm2.17$ | $5.07\pm3.57$ | $72.02\pm1.53$ | $7.01\pm0.37$ | $5.91\pm1.22$ |

*(a)* Performance with different values of $\tau$

| $\gamma$ | Wikipedia | | | MOOC | | |
|---|---|---|---|---|---|---|
| | AUROC | AP | F1 | AUROC | AP | F1 |
| 0.99 | $80.36\pm0.69$ | $5.41\pm1.23$ | $8.34\pm2.72$ | $72.87\pm1.40$ | $7.89\pm0.38$ | $7.15\pm0.64$ |
| 0.95 | $80.29\pm1.74$ | $5.28\pm1.76$ | $3.51\pm0.81$ | $72.57\pm0.48$ | $7.18\pm0.34$ | $7.02\pm0.80$ |
| 0.90 | $80.77\pm0.72$ | $5.57\pm1.69$ | $5.79\pm1.85$ | $72.98\pm0.95$ | $7.24\pm0.44$ | $6.02\pm0.29$ |
| 0.80 | $81.15\pm0.68$ | $5.43\pm1.70$ | $3.97\pm0.67$ | $72.63\pm0.87$ | $7.40\pm0.27$ | $6.65\pm0.66$ |

*(b)* Performance with different values of $\gamma$

*Table 4.* Effect of hyperparameters $\tau$ and $\gamma$ on performance on Wikipedia and MOOC datasets. The best results are highlighted in **green**, and the *underlined* entries correspond to the results reported in Table. 1.

for balancing precision and generalization. For $L$, AUROC gradually increases as $L$ increases, while F1 consistently decreases. This phenomenon aligns well with the characteristics of residual representation. When more historical information is aggregated during representation encoding, the residual signals become diluted, thereby reducing the discriminative capacity between normal and abnormal samples and directly lowering F1 performance. At the same time, AUROC remains less sensitive to this dilution because it evaluates ranking consistency rather than absolute separability. Even when anomaly and normal scores converge and exhibit weaker discriminability, as long as anomalies tend to be ranked above normal samples, AUROC will continue to increase. This explains why larger $L$ yields higher AUROC but lower F1. The results for hyperparameter $\alpha$ highlight the importance of selecting an appropriate boundary. A larger $\alpha$ pulls the normal boundary toward the center of the normal distribution, which typically improves robustness and generalization, but reduces discriminability by treating borderline normal samples as anomalous. In contrast, a smaller $\alpha$ yields a tighter boundary that can improve discriminability but increases the risk of overfitting. Overall, $\alpha = 0.01$ provides the most stable trade-off across datasets.

The results on datasets with synthetic anomalies are illustrated in Fig. 6 and Fig. 7, respectively. Our framework is insensitive to hyperparameter choices on these datasets. Performance changes slightly and the overall trend remains stable, indicating that SDGAD does not require delicate tuning to achieve strong results. This robustness is likely because synthetic anomalies exhibit clearer pattern shifts, so likelihood-space separation can be maintained even when the boundary or restriction strength varies moderately.

## 6. Conclusion

We propose SDGAD, an effective, generalizable and model-agnostic framework for dynamic graph anomaly detection. It learns anomaly-sensitive representations via residual encoding and stabilizes them with a restriction loss to improve separability under diverse anomaly patterns. And then explicitly learns an discriminative and robust boundary via a bi-boundary optimization strategy based on the normal log-likelihood distribution modeled by a normalizing flow. Extensive evaluations demonstrate its superiority. Discussion and future work are provided in Appendix A.

## Acknowledgements

The research presented in this paper is supported in part by National Natural Science Foundation of China (62372362).

## Impact Statement

This paper presents a work whose goal is to advance the field of Machine Learning. The potential societal consequence of our work is that it can help improve anomaly detection in real applications. However, we do not think that any specific societal impact needs to be highlighted here.

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

# A. Discussion and Future Work

In this section, we outline several practical directions for future work motivated by our empirical findings.

First, it is necessary to revisit how synthetic anomalies are constructed. Our results suggest that the synthetic anomaly protocols commonly used in prior work may be overly simple for some datasets. This issue is particularly clear on LastFM, where most baselines achieve similar performance with only a small number of labeled anomalies. A likely reason is that LastFM has a low rate of repeated edges, making anomalies generated by repeating edges at distant timestamps easy to detect. Future work should develop synthetic anomaly and evaluation protocols that account for dataset characteristics and produce anomalies whose detectability is not driven mainly by simple cues. More concretely, these protocols should control the salience of synthetic anomalies and support a more diverse set of anomaly patterns.

Second, future work should extend the problem formulation from binary detection to anomaly typing. In many applications, flagging an event as anomalous is insufficient; practitioners also need to understand the type of anomaly, since different underlying mechanisms require different responses. However, existing real-world datasets usually annotate anomalies as a single class, which prevents systematic analysis of type-level behavior. A natural next step is to enrich benchmarks with finer-grained annotations or to introduce evaluation settings in which models must separate anomalies into interpretable groups. From the modeling perspective, this direction calls for methods that can capture heterogeneity among anomalies and produce type-aware outputs, potentially with lightweight human verification when labels are scarce.

# B. Details of Datasets

| Dataset | Nodes | Edges | Unique Edges | Timesteps | Edge Feature | Anomaly Ratio | Density |
|---|---|---|---|---|---|---|---|
| Wikipedia | 9227 | 157474 | 18257 | 152757 | 172 | 0.14% | 4.30E-03 |
| Reddit | 10984 | 672447 | 78516 | 669065 | 172 | 0.05% | 8.51E-03 |
| MOOC | 7144 | 411749 | 178443 | 345600 | 4 | 0.99% | 1.26E-02 |
| LastFm | 1980 | 1293103 | 154993 | 1283614 | 0 | - | 5.57E-01 |
| Enron | 184 | 125235 | 3125 | 22632 | 0 | - | 5.53E+00 |
| UCI | 1899 | 59835 | 20296 | 58911 | 0 | - | 3.66E-02 |

*Table 5.* Dataset statistics

## B.1. Description of Datasets

**Wikipedia**: A bipartite graph of user edits on Wikipedia pages, where nodes represent users and pages, and edges denote timestamped edits. Each interaction is associated with a 172-dimensional LIWC feature. Dynamic labels indicate whether the corresponding edit behavior is banned.

**Reddit**: A bipartite dataset of user posts on Reddit over one month. Nodes correspond to users and subreddits, with timestamped posting edges and 172-dimensional LIWC features. Dynamic labels indicate whether the corresponding post behavior is banned.

**MOOC**: A bipartite interaction network between students and course units (e.g., videos, problem sets). Edges represent student access behaviors with 4-dimensional features. Dynamic labels indicate whether the access behavior is banned.

**LastFM**: A bipartite graph of music listening activities over one month, where nodes are users and songs, and edges denote timestamped listening events.

**Enron**: An email communication network with about 50K messages exchanged among Enron employees over three years. No attributes are provided.

**UCI**: A communication network among students at the University of California, Irvine, with timestamped interactions at second-level granularity. No additional features are included.

## B.2. Data Preprocessing

**Dataset Split Setup.** We split all datasets into three chronological segments for training, validation, and testing with ratios of 40%-20%-40%. For the three datasets with real anomalies, the anomaly proportion remains relatively consistent across

the training, validation, and testing sets. Specifically, the anomaly ratios are 0.14%, 0.15%, and 0.13% for Wikipedia, 1.14%, 0.86%, and 0.90% for MOOC, and 0.024%, 0.065%, and 0.08% for Reddit across the training, validation, and test set, respectively.

**Synthetic Anomaly Injection Protocol.** For the three datasets (LastFM, Enron, and UCI) that do not contain real anomalies, we inject synthetic anomalies following the dynamic graph anomaly synthesis protocol of Postuvan et al. (2024). Specifically, Postuvan et al. (2024) propose five synthesis strategies derived from three primitive anomaly types: (i) randomizing destination nodes to create structural anomalies, (ii) randomizing attributes to create contextual anomalies, and (iii) randomizing edge timestamps to create temporal anomalies. Because these datasets lack original node/edge attributes, type (ii) is not applicable. We therefore use only (i) and (iii) and construct two synthetic anomaly types: T (temporal), S (structural). To rigorously evaluate generalization across anomaly types, we inject only T anomalies into the training and validation sets at a rate of 0.1%, and evaluate on a test set with rarer and more diverse anomalies by injecting 0.05% anomalies of each type (T, S), where S are unseen during training.

## C. Baselines and Implementation Details

Our experiments cover both widely used discrete-time dynamic graph (DTDG)-based DGAD baselines and recent continuous-time dynamic graph (CTDG)-based DGAD methods. Since CTDG-based DGAD remains relatively under-explored with few dedicated methods, we follow prior works (Tian et al., 2023; Yang et al., 2025; Postuvan et al., 2024) and additionally include some representative CTDG models originally developed for fundamental dynamic graph task (e.g., link prediction), which can be directly adapted to anomaly detection.

**Implementation Details.** Following Postuvan et al. (2024), we apply a minimal modification when adapting general CTDG models: unlike standard link prediction that excludes the current target interaction from the input history, anomaly detection can include it. This adjustment allows CTDG models to operate consistently under the anomaly detection setting and ensures fair comparisons. All baselines are trained for up to 200 epochs with early stopping (patience = 10) and the checkpoint achieving the best validation performance is used for testing. Validation performance is also evaluated using AUROC, AP, and F1. Since computing F1 requires a threshold $\epsilon$, we set $\epsilon = 0.5$ for all experiments during validation. Therefore, we report test-time F1 using the same threshold $\epsilon = 0.5$ for all experiments to ensure a consistent and fair evaluation. We fix the batch size to 200 for all methods and datasets. We report the average and standard deviation over five independent runs. We optimize model parameters using Adam (Kingma & Ba, 2015) and conduct grid search over the main Adam hyperparameters: the learning rate is selected from $\{1e-3, 1e-4, 1e-5\}$ and the weight decay from $\{1e-1, 1e-2, 1e-3, 1e-4, 1e-5\}$. We run all the experiments with NVIDIA GeForce RTX 4090 GPUs.

### C.1. DTDG-based DGAD Methods

**Netwalk** (Yu et al., 2018) A temporal random-walk embedding model that learns joint node–edge representations. It maintains an online clustering of embedding trajectories and flags deviations as anomalies.

**AddGraph** (Zheng et al., 2019) A semi-supervised DTDG method that augments a temporal GCN with attention to capture long- and short-term patterns, and trains with selective negative sampling plus a margin loss to address label sparsity.

**StrGNN** (Cai et al., 2021) A subgraph-based temporal model for edge anomalies. It extracts the $h$-hop enclosing subgraph, labels node roles, applies graph convolution with SortPooling to obtain fixed-size snapshot features, and uses a GRU to capture temporal dynamics.

**Taddy** (Liu et al., 2023) A dynamic graph transformer with a learnable node encoding that separates global spatial, local spatial, and temporal terms. It samples edge-centered temporal substructures and uses attention to couple structural and temporal dependencies end to end.

### C.2. CTDG-based DGAD Methods

**SAD** (Tian et al., 2023) A semi-supervised CTDG method for DGAD. It first predicts node-level anomaly scores and stores score–timestamp pairs in a memory bank to estimate a normal prior and apply a deviation loss, and adds a pseudo-label contrastive module that forms score-based pseudo-groups and treats intra-group pairs as positives.

**GeneralDyG** (Yang et al., 2025) An unsupervised CTDG method for DGAD. It uses a GNN extractor that embeds nodes,

edges, and topology and alternates node- and edge-centric message passing. It inserts special tokens into feature sequences to encode hierarchical relations between anomalous events while balancing global temporal context and local dynamics, and trains on ego-graph samples of anomalous events to reduce computation.

## C.3. Representative CTDG Models

**JODIE** (Kumar et al., 2019)    A memory-based CTDG model for interaction streams. It maintains a dynamic embedding per node and updates it only at observed interactions, using the node's previous embedding, the counterpart's current embedding, interaction features, and the time since the node's last interaction. For prediction at a later time, it applies a learned time-dependent projection to the most recent embedding.

**DyRep** (Trivedi et al., 2019)    A continuous-time CTDG model that integrates representation learning with a temporal point process. It maintains a latent state per node and updates it at event times by combining self-updating from the previous state with attention-based aggregation over recent neighbors. The updated state is used as the node representation, and node-pair states are further used to parameterize the point-process intensity for modeling event occurrence.

**TGN** (Rossi et al., 2020)    A modular memory-based CTDG model. It maintains a memory state for each node and updates it only when the node participates in an interaction. For each event, it constructs messages from the endpoints' current memories together with edge and time information, aggregates messages per node, and applies a recurrent updater to refresh the memories. The time-specific node representation is then computed from the updated memory, optionally enhanced by attention over a sampled temporal neighborhood.

**TGAT** (Xu et al., 2020)    A temporal graph attention model that computes node representations by attending over time-stamped neighbors. For a queried node at a queried time, it gathers historical neighbor interactions that happened earlier, and attaches a continuous time encoding to each interaction based on its recency. Multi-head attention then weights and combines these temporally encoded neighbor interactions to form the node representation.

**GraphMixer** (Cong et al., 2023)    A lightweight sequence model based on MLP-Mixer blocks. For a queried node and time, it builds an ordered sequence from the node's historical interactions, encodes each timestamp using a fixed cosine-based time encoding with pre-defined frequencies, and combines this with link information to form interaction tokens. Stacked MLP-Mixer blocks alternately mix information across the temporal positions and across feature channels.

**TCL** (Wang et al., 2021a)    A transformer-based CTDG model. For each interaction, it constructs temporal dependency subgraphs for the node pair and encodes them as ordered sequences with node, hierarchy, and time-gap information. A structure-aware transformer restricts attention to valid dependency contexts, and a cross-attention module exchanges information between the two endpoint histories to produce time-aware node representations.

**CAWN** (Wang et al., 2021c)    A walk-based CTDG model based on causal anonymous walks. For a given query node, it samples multiple time-respecting random walks that trace historical interactions, with a bias toward more recent edges. Instead of using raw node identities, it adopts an anonymous encoding derived from node visit statistics in the sampled walks. Each walk is then encoded as a short time-stamped sequence with a recurrent model, and the resulting walk embeddings are aggregated to form the final node representation.

**DyGFormer** (Yu et al., 2023)    A transformer-based CTDG model. For an interaction, it gathers the chronological first-hop histories of both endpoints and encodes each interaction using neighbor information, edge features, and a sinusoidal time-interval encoding. It further incorporates a co-occurrence signal to reflect shared neighbors across the two histories. To improve efficiency on long sequences, it groups consecutive interactions into patches and applies a transformer over the patch sequence.

**FreeDyG** (Tian et al., 2024b)    A frequency-aware CTDG model. It encodes a fixed-length temporal neighbor sequence using node, edge, and time information, and augments it with interaction and co-occurrence frequency signals. It then applies a frequency-enhanced MLP-Mixer that transforms the sequence to the frequency domain, filters informative components with learnable weights, and transforms back to the time domain for further mixing.

## D. Model Complexity Analysis

Let $B$ be the batch size, and let $T_{\text{enc}}(L)$ denote the cost of one CTDG encoder call on a node history of length $L$. For a baseline built on the same backbone, each event requires encoding the two endpoints once, so the per-batch cost is

$$T_{\text{base}} = O(2B\,T_{\text{enc}}(L))$$

In SDGAD, the main extra cost comes from residual construction in Eq. (3), where each endpoint is encoded once with the current event and once without it. This increases the encoder-side cost to $O(4B\,T_{\text{enc}}(L))$ The restriction module only adds a linear projection, with cost $O(B\,d_r d_x)$, where $d_r$ and $d_x$ denote the residual and projected dimensions.

The normalizing flow is applied after representation learning. In our implementation, the Jacobian determinant does not require forming or computing a full dense Jacobian at each forward pass. Instead, we use coupling-based invertible blocks, for which the log-determinant can be evaluated analytically from the scaling terms. In practice, we use 6 coupling blocks with two-layer fully connected subnetworks, resulting in a cost of $O(6B\,d_x^2)$. Therefore, although the flow does introduce additional computation, its overhead remains much smaller than the encoder cost $T_{\text{enc}}(L)$ in our setting. Therefore, the total per-batch complexity is

$$T_{\text{SDGAD}} = O\big(4B\,T_{\text{enc}}(L) + B\,d_r d_x + 6B\,d_x^2\big)$$

Overall, the dominant cost in our framework still comes from the CTDG encoder, while the additional modules introduce only relatively small overhead. ***Notably***, although the complexity formula suggests more than a $2\times$ increase, the two branches in residual construction are independent and can be executed in parallel. Therefore, the actual increase in inference time is much smaller.

## E. Loss Analysis

| $\lambda_1/\lambda_2$ | Wikipedia | | | MOOC | | |
|---|---|---|---|---|---|---|
| | AUROC | AP | F1 | AUROC | AP | F1 |
| 0.5/0.5 | $71.73_{\pm 8.80}$ | $4.08_{\pm 3.17}$ | $4.18_{\pm 3.93}$ | $\mathbf{72.87}_{\pm 1.40}$ | $\mathbf{7.89}_{\pm 0.38}$ | $7.15_{\pm 0.64}$ |
| 1/0.1 | $\mathbf{80.67}_{\pm 1.14}$ | $4.47_{\pm 0.74}$ | $8.32_{\pm 2.00}$ | $70.02_{\pm 1.15}$ | $5.22_{\pm 1.26}$ | $4.74_{\pm 4.92}$ |
| 1/0.5 | $80.36_{\pm 0.69}$ | $\mathbf{5.41}_{\pm 1.23}$ | $\mathbf{8.34}_{\pm 2.72}$ | $70.48_{\pm 0.76}$ | $5.37_{\pm 1.14}$ | $\mathbf{7.69}_{\pm 4.52}$ |
| 1/0.7 | $80.64_{\pm 0.47}$ | $5.33_{\pm 1.47}$ | $6.41_{\pm 3.08}$ | $66.10_{\pm 7.71}$ | $4.30_{\pm 2.74}$ | $4.90_{\pm 5.27}$ |
| 1/1 | $76.65_{\pm 9.03}$ | $4.71_{\pm 2.19}$ | $6.78_{\pm 4.27}$ | $70.87_{\pm 0.51}$ | $5.45_{\pm 0.73}$ | $6.24_{\pm 2.01}$ |
| 1/2 | $74.91_{\pm 8.65}$ | $4.01_{\pm 2.90}$ | $4.65_{\pm 4.48}$ | $70.75_{\pm 0.82}$ | $5.80_{\pm 1.54}$ | $6.62_{\pm 1.18}$ |

*Table 6.* Results when varying different $\lambda_1/\lambda_2$ values for balancing loss terms.

### E.1. Sensitivity of Balancing The Loss Terms

Our framework jointly optimizes three objectives: the maximum-likelihood loss $\mathcal{L}_{\mathcal{ML}}$, the restriction loss $\mathcal{L}RR$, and the bi-boundary optimization loss $\mathcal{L}_{\mathcal{BO}}$. Since $\mathcal{L}_{\mathcal{ML}}$ is the standard training objective for normalizing flows, we fix its weight to 1 and use $\lambda_1$ and $\lambda_2$ to control the contributions of $\mathcal{L}_{\mathcal{BO}}$ and $\mathcal{L}_{\mathcal{RR}}$, respectively. Table 6 reports the results under different coefficient settings. Overall, SDGAD is robust to moderate variations in these weights and does not rely on fine-grained tuning. Across both datasets, $\lambda_2 = 0.5$ consistently yields the best performance, suggesting that a moderate restriction strength provides a good balance between compactness and preserving informative deviations in the residual space. In contrast, the optimal weight of $\mathcal{L}_{\mathcal{BO}}$ is dataset-dependent, which is expected given that anomaly characteristics differ substantially across domains. On Wikipedia, a stronger bi-boundary term is beneficial: reducing its weight to $\lambda_1 = 0.5$ leads to clear degradation, while $\lambda_1 = 1$ consistently achieves higher AUROC and F1. On MOOC, however, a lighter $\mathcal{L}_{\mathcal{BO}}$ term ($\lambda_1 = 0.5$) performs best, indicating that overly aggressive margin enforcement may harm generalization.

### E.2. Error Bound Analysis

#### E.2.1. ANALYSIS FOR $\mathcal{L}_{\mathcal{ML}}$ AND $\mathcal{L}_{\mathcal{BO}}$

We first focus on two loss terms, $\mathcal{L}_{\mathcal{ML}}$ and $\mathcal{L}_{\mathcal{BO}}$, since both are associated with the normalizing flow module. In contrast, $\mathcal{L}_{\mathcal{RR}}$ is optimized in a separate module and does not directly affect the error bound characterized in Proposition 1.

**Proposition 1.** *Assume that $\boldsymbol{\Phi}_{\theta^*} \in \mathrm{argmin}_{\theta \in \Theta}\{\mathcal{L}_{\mathcal{ML}} + \lambda_1 \mathcal{L}_{\mathcal{BO}}\}$. That is, $\boldsymbol{\Phi}_{\theta^*}$ corresponds to the optimal parameters of the normalizing flow module that minimize the joint objective of the maximum-likelihood loss $\mathcal{L}_{\mathcal{ML}}$ and the bi-boundary optimization loss $\mathcal{L}_{\mathcal{BO}}$. Then we have:*

$$\mathbb{E}_{y_i=0}[\max((\mathcal{B}'_n - \log[p(x_i)]), 0)] + \mathbb{E}_{y_j=1}[\max((\log[p(x_j)] - \mathcal{B}'_a), 0)]$$

$$\leq (\mathcal{B}_n - \mathcal{B}_a)\mathcal{L}_{\mathcal{BO}}(\boldsymbol{\Phi}_{\theta^*}) + \frac{N}{(N+M)}[\max(1 + \mathcal{B}'_n, -\mathcal{B}'_a)]$$

$$\leq \frac{(\frac{d}{2}\log(2\pi) - \frac{1}{2})(\mathcal{B}_n - \mathcal{B}_a)}{\lambda_1} + \frac{N}{(N+M)} \tag{14}$$

*where $y = 0$, $y = 1$ denote normal and anomalous labels, $\mathcal{B}'_n = \mathcal{B}_n - \epsilon$, $\mathcal{B}'_a = \mathcal{B}_a + \epsilon$, $\epsilon \in (0, \mathcal{B}_n - \mathcal{B}_a)$, $N$ and $M$ are the number of normal and abnormal samples.*

***proof.*** Suppose that we can sort all samples (both normal and anomalous) by their log-likelihood values in descending order: $\log[p(x_1)] \geq \log[p(x_2)] \geq \cdots \geq \log[p(x_{N+M})]$. Let $\mathcal{B}_n = \log[p(x_N)]$ denote the normal boundary induced by the $N$-th ranked sample, which corresponds to the threshold for classifying normal samples. Under a worst-case assumption, all top-$N$ samples which ideally should be normal are misclassified, while the remaining $M$ anomalous samples have log-likelihoods lying between $\mathcal{B}_a$ and $\mathcal{B}_n$. In this scenario, the expected margin-violation error can be bounded as follow:

$$\mathbb{E}_{y_i=0}[\max((\mathcal{B}'_n - \log[p(x_i)]), 0)] + \mathbb{E}_{y_j=1}[\max((\log[p(x_j)] - \mathcal{B}'_a), 0)]$$

$$\leq (\mathcal{B}'_n - \mathcal{B}'_a)\mathcal{L}'_{\mathcal{BO}}(\boldsymbol{\Phi}_{\theta^*}) + \frac{N}{(N+M)}[\max(1 + \mathcal{B}'_n, -\mathcal{B}'_a)]$$

$$\leq (\mathcal{B}_n - \mathcal{B}_a)\mathcal{L}_{\mathcal{BO}}(\boldsymbol{\Phi}_{\theta^*}) + \frac{N}{(N+M)} \tag{15}$$

Here $\mathcal{L}'_{\mathcal{BO}}$ denotes the $\ell_0$ norm based formulation of $\mathcal{L}_{\mathcal{BO}}$, which counts the number of samples violating the boundary constraints (i.e., the number of misclassified samples). It represents an idealized, non-differentiable version of $\mathcal{L}_{\mathcal{BO}}$, used only for theoretical analysis. The second inequality is obtained as $1 + \mathcal{B}'_n \leq 1$ and $-\mathcal{B}'_a \leq 1$ when $-1 \leq \mathcal{B}'_a < \mathcal{B}'_n \leq 0$ satisfies. Since $\boldsymbol{\Phi}_{\theta^*}$ is defined as the optimal parameter of the joint objective $\mathcal{L}_{\mathcal{ML}} + \lambda_1 \mathcal{L}_{\mathcal{BO}}$, its objective value cannot be larger than that of any other candidate solution. In particular, consider an arbitrary reference solution $\boldsymbol{\Phi}_{\theta'}$ such that $\mathcal{L}_{\mathcal{BO}}(\boldsymbol{\Phi}_{\theta'}) = 0$. By the optimality of $\boldsymbol{\Phi}_{\theta^*}$, we have:

$$\mathcal{L}_{\mathcal{ML}}(\boldsymbol{\Phi}_{\theta^*}) + \lambda_1 \mathcal{L}_{\mathcal{BO}}(\boldsymbol{\Phi}_{\theta^*}) \leq \mathcal{L}_{\mathcal{ML}}(\boldsymbol{\Phi}_{\theta'}) + \lambda_1 \mathcal{L}_{\mathcal{BO}}(\boldsymbol{\Phi}_{\theta'})$$

$$= \mathcal{L}_{\mathcal{ML}}(\boldsymbol{\Phi}_{\theta'}) \tag{16}$$

We isolate $\mathcal{L}_{\mathcal{BO}}(\boldsymbol{\Phi}_{\theta^*})$ as:

$$\mathcal{L}_{\mathcal{BO}}(\boldsymbol{\Phi}_{\theta^*}) \leq \frac{(\mathcal{L}_{\mathcal{ML}}(\boldsymbol{\Phi}_{\theta'}) - \mathcal{L}_{\mathcal{ML}}(\boldsymbol{\Phi}_{\theta^*}))}{\lambda_1}$$

$$\leq \frac{\left(\frac{1}{2}\boldsymbol{\Phi}_{\theta'}(x)^T\boldsymbol{\Phi}_{\theta'}(x) - \frac{1}{2}\boldsymbol{\Phi}_{\theta^*}(x)^T\boldsymbol{\Phi}_{\theta^*}(x) + \frac{1}{2}\boldsymbol{\Phi}_{\theta^*}(x)^T\boldsymbol{\Phi}_{\theta^*}(x) + \frac{d}{2}\log(2\pi)\right)}{\lambda_1}$$

$$\leq \frac{\frac{d}{2}\log(2\pi) - \frac{1}{2}}{\lambda_1} \tag{17}$$

To obtain a tractable bound, we assume a worst-case initialization:

$$\boldsymbol{\Phi}_{\theta'}(x)^T\boldsymbol{\Phi}_{\theta'}(x) = -1 \tag{18}$$

This assumption gives the largest possible gap between $\boldsymbol{\Phi}_{\theta'}$ and $\boldsymbol{\Phi}_{\theta^*}$, and thus produces the loosest valid bound. By combining the above E.q.(15) and E.q.(17), we have that

$$\mathbb{E}_{y_i=0}[\max((\mathcal{B}'_n - \log[p(x_i)]), 0)] + \mathbb{E}_{y_j=1}[\max((\log[p(x_j)] - \mathcal{B}'_a), 0)]$$

$$\leq \frac{(\frac{d}{2}\log(2\pi) - \frac{1}{2})(\mathcal{B}_n - \mathcal{B}_a)}{\lambda_1} + \frac{N}{(N+M)} \tag{19}$$

The above proposition highlights both the necessity and the effectiveness of the bi-boundary optimization loss $\mathcal{L}_{\mathcal{BO}}$. Ideally, increasing the weight $\lambda_1$ of $\mathcal{L}_{\mathcal{BO}}$ facilitates the convergence of the error bound toward zero. Moreover, the proposition implies that the presence of more anomalous samples (larger $M$) can further enhance the reliability of discriminating between normal and abnormal samples.

### E.2.2. ANALYSIS FOR $\mathcal{L}_{\mathcal{RR}}$

We now provide the error bound analysis for the representation restriction loss $\mathcal{L}_{RR}$.

**Proposition 2.** Assume that the induced pseudo-Huber norm satisfies $0 \leq n(x) \leq 1$ for all samples.[2] Then we can define the expected margin-violation error $\mathcal{E}_{RR}$ as

$$\mathcal{E}_{RR} = \mathbb{E}_{y=0}[\max(r_{\min} - n(x), 0) + \max(n(x) - r_{\max}, 0)] + \mathbb{E}_{y=1}[\max(r' - n(x), 0)] \tag{20}$$

Then,

$$\mathcal{E}_{RR} \leq \frac{C_r}{\log 2} \qquad C_r = \max\{r_{\min}, 1 - r_{\max}, r'\} \tag{21}$$

*proof.* For a normal sample ($y = 0$), if $n(x) \in [r_{\min}, r_{\max}]$, then both the violation term and $L_n(x)$ are zero. Otherwise, let $\delta = r_{\min} - n(x) > 0$ when $n(x) \leq r_{\min}$, or $\delta = n(x) - r_{\max} > 0$ when $n(x) \geq r_{\max}$, in both cases

$$L_n(x) = -\sigma(-\delta) e^{\delta} = \log(1 + e^{\delta}) e^{\delta} \geq \log 2 \tag{22}$$

which implies $\mathbb{I}[n(x) \notin [r_{\min}, r_{\max}]] \leq L_n(x)/\log 2$. Under $0 \leq n(x) \leq 1$, the normal violation magnitude is upper bounded by

$$\max(r_{\min} - n(x), 0) + \max(n(x) - r_{\max}, 0) \leq \max\{r_{\min}, 1 - r_{\max}\} \cdot \mathbb{I}[n(x) \notin [r_{\min}, r_{\max}]] \leq \frac{\max\{r_{\min}, 1 - r_{\max}\}}{\log 2} L_n(x) \tag{23}$$

For an anomalous sample ($y = 1$), if $n(x) > r'$, then both the violation term and $L_a(x)$ are zero. Otherwise let $\delta = r' - n(x) > 0$,

$$L_a(x) = \log(1 + e^{\delta}) e^{\delta} \geq \log 2 \quad \Rightarrow \quad \mathbb{I}[n(x) \leq r'] \leq L_a(x)/\log 2 \tag{24}$$

Using $n(x) \geq 0$, the anomalous violation magnitude satisfies

$$\max(r' - n(x), 0) \leq r' \cdot \mathbb{I}[n(x) \leq r'] \leq \frac{r'}{\log 2} L_a(x). \tag{25}$$

Taking expectations over $y = 0$ and $y = 1$ and combining the two bounds yields

$$\mathcal{E}_{RR} \leq \frac{C_r}{\log 2} \left( \mathbb{E}_{y=0}[L_n(x)] + \mathbb{E}_{y=1}[L_a(x)] \right) = \frac{C_r}{\log 2} \tag{26}$$

Because $L_n(x) = 0$ once $n(x) \in [r_{\min}, r_{\max}]$ and $L_a(x) = 0$ once $n(x) > r'$, minimizing $\mathcal{L}_{RR}$ only suppresses the failure mode where samples cross the hypersphere constraints. It does not further contract already-feasible normals nor does it force already-separated anomalies toward a single norm scale. This is desirable under heterogeneous anomaly types with scale variance, since the restriction acts as a feasibility regularizer rather than collapsing diverse anomalies into a fixed class.

## F. Characteristics of Evaluation Metrics under Extreme Class Imbalance

### F.1. Why AUROC can be overly optimistic

In anomaly detection task under extreme class imbalance, negative (normal) samples typically dominate the dataset, far exceeding the number of positive (anomalous) samples. Under this setting, ROC-based evaluation (AUROC) can yield an overly optimistic assessment. This follows from the definition of the ROC x-axis, the false positive rate (FPR), which normalizes false positives by the total number of negatives:

$$\text{TPR} = \frac{\text{TP}}{\text{TP} + \text{FN}}, \quad \text{FPR} = \frac{\text{FP}}{\text{FP} + \text{TN}} \tag{27}$$

---

[2]This boundedness is consistent with the unit-sphere scaling used to set $r_{\max}$.

Here TP, FP, TN, and FN denote entries of the confusion matrix. When TN is very large, even a substantial absolute increase in FP results in only a small change in FPR. Consequently, the ROC curve (and thus AUROC) can remain high despite a large number of false alarms in absolute terms. By contrast, precision explicitly penalizes false positives relative to true positives:

$$\text{Precision} = \frac{\text{TP}}{\text{TP} + \text{FP}}, \quad \text{Recall} = \frac{\text{TP}}{\text{TP} + \text{FN}} \tag{28}$$

and therefore more directly reflects the impact of false positives under skewed class distributions. The same mechanism also explains why AUROC can remain high even when anomaly scores collapse into a narrow, low-valued range. If the score distributions overlap heavily but preserve a weak ordering signal (i.e., anomalies are only marginally higher than normals), many thresholds may still yield ROC points with relatively high TPR at low FPR, because FPR varies slowly when dominated by TN. Under the same conditions, AUPRC and threshold-dependent metrics typically degrade more noticeably, as they are more sensitive to FP through Precision.

### F.2. Metric trade-offs in hyper-parameter selection

Under extreme class imbalance, AUROC, AUPRC, and F1 reflect different aspects of score quality and thus are generally not optimized by the same hyperparameter configuration. AUROC and AP both summarize performance by sweeping decision thresholds, but they operate in different spaces: AUROC is defined in ROC space, where $\text{FPR} = \text{FP}/(\text{FP} + \text{TN})$ is normalized by the typically large TN and can therefore be relatively insensitive to accumulated false positives, whereas AP is defined in PR space and penalizes false positives directly through $\text{Precision} = \text{TP}/(\text{TP} + \text{FP})$. In contrast, F1 is evaluated at a single operating threshold, making it sensitive to how well anomalies are separated in the high-score region and to threshold-induced changes in FP and Precision. Consequently, a configuration that improves AUROC does not necessarily improve AP, and may fail to yield the best F1 under practical thresholding rules.

## G. Thresholds Analysis

In this section, we explore on the relationship among F1, TPR, and FPR under different thresholds. We conduct experiments using SDGAD (TCL) and TCL on the Wikipedia and UCI datasets. During validation, F1 was still computed using the fixed threshold of 0.5. On the test set, we report F1, TPR, and FPR under different thresholds, specifically 0.01, 0.1, 0.3, and 0.5. The corresponding results are presented in the Tables 7, 8, 9, 10.

It can be observed that, on the relatively simple synthetic anomaly dataset uci, TCL yields $F1 = 0$, $\text{TPR} = 0$, and $\text{FPR} = 0$ under all tested thresholds. However, SDGAD consistently achieves $F1 = 0.4444$, $\text{TPR} = 1$, and $\text{FPR} = 0.002298$ across all tested thresholds. This suggests that, for unsupervised methods, manually specifying a good threshold for anomaly detection is of very limited use. In this case, all scores produced by TCL are even below 0.01, making threshold-based discrimination ineffective. By contrast, our method substantially enlarges the score range, leading to much better separability and a clear distinction between anomalous and normal samples.

On the more challenging real-world anomaly dataset, although the effect is less pronounced than on UCI, a similar pattern can still be observed. Although TCL yields a low FPR at all tested thresholds, and its FPR further decreases as the threshold increases, its TPR also drops steadily. As a result, its overall F1 score remains poor. This indicates that TCL is not sensitive to threshold changes and provides limited class separability. In contrast, our method responds much more clearly to threshold variation: as the threshold increases, both TPR and FPR change substantially. This behavior indicates that our method achieves much better separability between anomalous and normal samples. Although our method has a higher absolute FPR than TCL, the trade-off between TPR and FPR is consistently more favorable for our method.

In addition, we clarify two possible concerns. (1) Under a very low threshold (e.g., $\epsilon = 0.01$), our method yields an FPR close to 1. This is expected because our method produces a wider score range: as the threshold decreases, more samples are predicted as positive, leading to a higher FPR. This shows that the scores provide a usable operating range instead of collapsing to near-all-negative predictions. (2) The analysis uses the checkpoint selected by validation F1 at a fixed threshold of 0.5 and then evaluated on the test set. Using other validation thresholds for model selection may change the exact test-time relations among F1, TPR, and FPR, but the main conclusion remains unchanged: our method improves separability and offers a more usable threshold range for distinguishing anomalous from normal samples.

| Threshold | F1 | TPR | FPR |
|---|---|---|---|
| 0.01 | 0 | 0 | 0 |
| 0.1 | 0 | 0 | 0 |
| 0.3 | 0 | 0 | 0 |
| 0.5 | 0 | 0 | 0 |

*Table 7.* TCL on UCI

| Threshold | F1 | TPR | FPR |
|---|---|---|---|
| 0.01 | 0.444444 | 1 | 0.002298 |
| 0.1 | 0.444444 | 1 | 0.002298 |
| 0.3 | 0.444444 | 1 | 0.002298 |
| 0.5 | 0.444444 | 1 | 0.002298 |

*Table 8.* SDGAD on UCI

| Threshold | F1 | TPR | FPR |
|---|---|---|---|
| 0.01 | 0.029557 | 0.073171 | 0.000555 |
| 0.1 | 0.032520 | 0.024390 | 0.000620 |
| 0.3 | 0.020408 | 0.012195 | 0.000238 |
| 0.5 | 0.021277 | 0.012195 | 0.000175 |

*Table 9.* TCL on Wiki

| Threshold | F1 | TPR | FPR |
|---|---|---|---|
| 0.01 | 0.002664 | 1 | 0.976140 |
| 0.1 | 0.006867 | 0.695122 | 0.261700 |
| 0.3 | 0.031718 | 0.439024 | 0.034209 |
| 0.5 | 0.111111 | 0.097561 | 0.000858 |

*Table 10.* SDGADG on Wiki

# H. Additional Experiments

| Methods | UCI | | | Enron | | | LastFM | | |
|---|---|---|---|---|---|---|---|---|---|
| | AUROC | AP | F1 | AUROC | AP | F1 | AUROC | AP | F1 |
| CTDG-based DGAD Methods | | | | | | | | | |
| SAD | $96.27_{\pm3.11}$ | $21.59_{\pm4.91}$ | $12.14_{\pm3.48}$ | $92.47_{\pm2.16}$ | $12.33_{\pm2.83}$ | $8.91_{\pm3.59}$ | $99.88_{\pm0.01}$ | $31.47_{\pm1.27}$ | $44.86_{\pm0.52}$ |
| GeneralDyG | $78.19_{\pm10.20}$ | $8.78_{\pm7.70}$ | $4.57_{\pm10.22}$ | $86.94_{\pm3.91}$ | $7.00_{\pm3.45}$ | $0_{\pm0}$ | $99.85_{\pm0.03}$ | $29.34_{\pm0.16}$ | $44.61_{\pm0.76}$ |
| Representative CTDG Models | | | | | | | | | |
| JODIE | $42.26_{\pm3.26}$ | $0.40_{\pm0.15}$ | $0.28_{\pm0.63}$ | $44.83_{\pm1.32}$ | $2.90_{\pm2.01}$ | $0_{\pm0}$ | $97.94_{\pm0.61}$ | $27.18_{\pm0.49}$ | $42.08_{\pm1.19}$ |
| DyRep | $62.42_{\pm13.94}$ | $0.49_{\pm0.18}$ | $0_{\pm0}$ | $60.06_{\pm0.74}$ | $7.29_{\pm3.17}$ | $4.16_{\pm2.96}$ | $99.84_{\pm0.73}$ | $28.19_{\pm1.06}$ | $43.18_{\pm1.58}$ |
| TGN | $87.84_{\pm3.77}$ | $15.55_{\pm5.38}$ | $0_{\pm0}$ | $79.79_{\pm2.82}$ | $6.15_{\pm2.14}$ | $0_{\pm0}$ | $99.88_{\pm0.71}$ | $29.86_{\pm1.20}$ | $44.91_{\pm1.07}$ |
| TGAT | $93.06_{\pm2.53}$ | $18.88_{\pm11.71}$ | $0_{\pm0}$ | $85.78_{\pm5.18}$ | $6.23_{\pm2.17}$ | $0_{\pm0}$ | $99.88_{\pm0.01}$ | $29.53_{\pm0.01}$ | $44.54_{\pm0.11}$ |
| TCL | $81.84_{\pm9.20}$ | $3.34_{\pm0.59}$ | $0_{\pm0}$ | $70.63_{\pm1.53}$ | $16.26_{\pm2.72}$ | $1.73_{\pm1.54}$ | $99.88_{\pm0.01}$ | $29.49_{\pm0.04}$ | $44.85_{\pm0.16}$ |
| CAWN | $78.67_{\pm11.45}$ | $7.21_{\pm4.00}$ | $0_{\pm0}$ | $86.78_{\pm2.05}$ | $5.89_{\pm4.17}$ | $0_{\pm0}$ | $99.88_{\pm0.01}$ | $29.75_{\pm0.65}$ | $44.89_{\pm0.06}$ |
| GraphMixer | $85.47_{\pm5.34}$ | $0.33_{\pm4.75}$ | $0_{\pm0}$ | $71.01_{\pm12.50}$ | $0.47_{\pm0.31}$ | $0_{\pm0}$ | $99.02_{\pm0.33}$ | $28.64_{\pm1.21}$ | $42.81_{\pm0.45}$ |
| DyGFormer | $78.07_{\pm2.73}$ | $11.53_{\pm6.23}$ | $15.37_{\pm16.59}$ | $86.36_{\pm11.27}$ | $22.93_{\pm6.32}$ | $32.10_{\pm8.14}$ | $99.90_{\pm0.02}$ | $35.48_{\pm5.31}$ | $45.67_{\pm0.31}$ |
| FreeDyG | $82.02_{\pm5.01}$ | $1.57_{\pm4.83}$ | $0_{\pm0}$ | $76.32_{\pm7.12}$ | $3.20_{\pm2.11}$ | $0_{\pm0}$ | $99.16_{\pm0.04}$ | $29.17_{\pm0.84}$ | $43.11_{\pm0.45}$ |
| Ours | | | | | | | | | |
| SDGAD (TCL) | $\mathbf{99.93_{\pm0.00}}$ | $\mathbf{38.90_{\pm0.00}}$ | $\mathbf{56.00_{\pm0.00}}$ | $99.86_{\pm0.02}$ | $29.05_{\pm0.40}$ | $\mathbf{44.65_{\pm0.02}}$ | $99.91_{\pm0.91}$ | $29.52_{\pm0.06}$ | $45.73_{\pm0.10}$ |
| SDGAD (DyGFormer) | $99.88_{\pm0.01}$ | $28.66_{\pm0.49}$ | $24.39_{\pm0.20}$ | $\mathbf{99.89_{\pm0.01}}$ | $\mathbf{29.10_{\pm2.88}}$ | $40.59_{\pm2.36}$ | $\mathbf{99.92_{\pm0.04}}$ | $\mathbf{40.01_{\pm3.11}}$ | $\mathbf{45.89_{\pm0.33}}$ |
| SDGAD (CAWN) | $99.89_{\pm0.01}$ | $28.84_{\pm0.99}$ | $44.44_{\pm0.00}$ | $99.87_{\pm0.00}$ | $28.73_{\pm0.00}$ | $44.64_{\pm0.00}$ | $99.91_{\pm0.01}$ | $30.06_{\pm0.90}$ | $45.02_{\pm0.18}$ |

*Table 11.* Performance comparison ($Avg_{\pm std}$ in %) on datasets with synthetic anomalies under **S1**. Best results are highlighted in **green**.

| Methods | Few-shot=1 | | | Few-shot=2 | | | Few-shot=3 | | |
|---|---|---|---|---|---|---|---|---|---|
| | AUROC | AP | F1 | AUROC | AP | F1 | AUROC | AP | F1 |
| SAD | $64.11_{\pm4.20}$ | $1.03_{\pm0.24}$ | $0_{\pm0}$ | $63.49_{\pm3.18}$ | $0.74_{\pm0.17}$ | $0_{\pm0}$ | $64.56_{\pm3.07}$ | $0.90_{\pm0.08}$ | $0_{\pm0}$ |
| GeneralDyG | $63.02_{\pm5.01}$ | $0.26_{\pm0.06}$ | $0_{\pm0}$ | $63.83_{\pm5.45}$ | $0.28_{\pm0.07}$ | $0_{\pm0}$ | $64.73_{\pm3.47}$ | $0.29_{\pm0.05}$ | $0_{\pm0}$ |
| JODIE | $80.96_{\pm2.87}$ | $0.63_{\pm0.03}$ | $0_{\pm0}$ | $81.00_{\pm2.75}$ | $0.86_{\pm0.23}$ | $0_{\pm0}$ | $81.42_{\pm2.25}$ | $0.64_{\pm0.04}$ | $0_{\pm0}$ |
| +ours | $81.15_{\pm2.63}$ | $0.67_{\pm0.05}$ | $1.05_{\pm0.19}$ | $81.86_{\pm2.97}$ | $0.87_{\pm0.05}$ | $1.06_{\pm0.19}$ | $82.23_{\pm2.64}$ | $0.73_{\pm0.05}$ | $1.51_{\pm0.24}$ |
| DyRep | $80.70_{\pm1.86}$ | $0.66_{\pm0.05}$ | $0_{\pm0}$ | $80.64_{\pm1.75}$ | $0.66_{\pm0.05}$ | $0_{\pm0}$ | $82.05_{\pm2.22}$ | $0.68_{\pm0.06}$ | $0_{\pm0}$ |
| TGN | $83.01_{\pm3.04}$ | $1.46_{\pm0.32}$ | $0_{\pm0}$ | $82.13_{\pm2.43}$ | $0.87_{\pm0.11}$ | $0_{\pm0}$ | $83.39_{\pm2.50}$ | $0.89_{\pm0.14}$ | $0_{\pm0}$ |
| TAGT | $56.30_{\pm2.43}$ | $1.81_{\pm0.20}$ | $0_{\pm0}$ | $58.50_{\pm1.97}$ | $0.86_{\pm0.23}$ | $0_{\pm0}$ | $59.18_{\pm3.60}$ | $0.87_{\pm0.25}$ | $0_{\pm0}$ |
| +ours | $65.94_{\pm0.27}$ | $1.91_{\pm0.07}$ | $0.49_{\pm0.12}$ | $64.67_{\pm2.56}$ | $0.97_{\pm0.10}$ | $0.89_{\pm0.56}$ | $65.31_{\pm2.56}$ | $1.45_{\pm0.08}$ | $0.66_{\pm0.16}$ |
| GraphMixer | $75.29_{\pm0.30}$ | $0.50_{\pm0.19}$ | $0_{\pm0}$ | $73.69_{\pm2.84}$ | $0.51_{\pm0.18}$ | $0_{\pm0}$ | $70.43_{\pm5.03}$ | $0.63_{\pm0.37}$ | $0_{\pm0}$ |
| CAWN | $67.35_{\pm1.09}$ | $0.40_{\pm0.03}$ | $0_{\pm0}$ | $74.76_{\pm2.54}$ | $0.80_{\pm0.05}$ | $0_{\pm0}$ | $74.28_{\pm2.04}$ | $0.78_{\pm0.05}$ | $0_{\pm0}$ |
| DyGFormer | $77.81_{\pm2.24}$ | $0.52_{\pm0.11}$ | $0_{\pm0}$ | $79.14_{\pm1.65}$ | $0.48_{\pm0.05}$ | $0_{\pm0}$ | $79.20_{\pm1.62}$ | $0.50_{\pm0.14}$ | $0_{\pm0}$ |
| FreeDyG | $77.90_{\pm1.81}$ | $0.51_{\pm0.21}$ | $0_{\pm0}$ | $76.22_{\pm1.94}$ | $0.50_{\pm0.11}$ | $0_{\pm0}$ | $75.73_{\pm2.31}$ | $0.61_{\pm0.42}$ | $0_{\pm0}$ |
| TCL | $66.75_{\pm4.37}$ | $0.75_{\pm0.41}$ | $0_{\pm0}$ | $70.40_{\pm2.79}$ | $0.82_{\pm0.44}$ | $0_{\pm0}$ | $69.37_{\pm4.81}$ | $0.78_{\pm0.18}$ | $0_{\pm0}$ |
| +ours | $68.29_{\pm1.87}$ | $0.91_{\pm0.21}$ | $0_{\pm0}$ | $71.53_{\pm1.21}$ | $0.85_{\pm0.32}$ | $0_{\pm0}$ | $72.09_{\pm1.68}$ | $0.91_{\pm0.18}$ | $0.94_{\pm0.89}$ |

*Table 12.* Performance comparison ($Avg_{\pm std}$ in %) on Wikipedia dataset under **S2**.

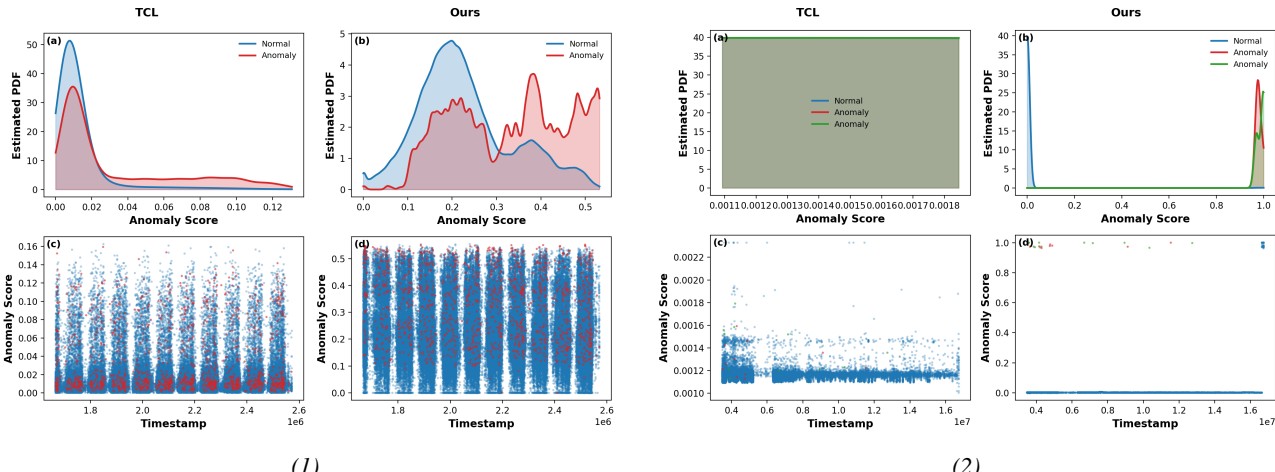

*Figure 5.* Visualization of anomaly score distributions on the MOOC (1) and UCI (2) test sets, respectively.

| Methods | Few-shot=1 | | | Few-shot=2 | | | Few-shot=3 | | |
|---|---|---|---|---|---|---|---|---|---|
| | AUROC | AP | F1 | AUROC | AP | F1 | AUROC | AP | F1 |
| SAD | $65.17_{\pm1.11}$ | $2.30_{\pm1.39}$ | $0_{\pm0}$ | $65.28_{\pm1.03}$ | $2.22_{\pm1.40}$ | $0_{\pm0}$ | $66.28_{\pm1.24}$ | $2.30_{\pm1.39}$ | $0_{\pm0}$ |
| GeneralDyG | $64.49_{\pm0.07}$ | $1.89_{\pm0.04}$ | $0_{\pm0}$ | $65.52_{\pm0.14}$ | $1.43_{\pm0.17}$ | $0_{\pm0}$ | $66.20_{\pm0.15}$ | $2.21_{\pm0.05}$ | $0_{\pm0}$ |
| JODIE | $57.34_{\pm4.61}$ | $1.39_{\pm0.20}$ | $0_{\pm0}$ | $57.91_{\pm4.45}$ | $1.43_{\pm0.17}$ | $0_{\pm0}$ | $58.32_{\pm4.66}$ | $1.43_{\pm0.19}$ | $0_{\pm0}$ |
| +ours | $60.21_{\pm1.09}$ | $1.48_{\pm0.03}$ | $3.20_{\pm0.18}$ | $60.68_{\pm1.21}$ | $1.49_{\pm0.02}$ | $3.23_{\pm0.16}$ | $61.35_{\pm1.02}$ | $1.67_{\pm0.03}$ | $3.25_{\pm0.19}$ |
| DyRep | $63.31_{\pm0.93}$ | $1.65_{\pm0.07}$ | $0_{\pm0}$ | $63.92_{\pm0.73}$ | $1.69_{\pm0.08}$ | $0_{\pm0}$ | $63.73_{\pm0.75}$ | $1.68_{\pm0.07}$ | $0_{\pm0}$ |
| TGN | $64.80_{\pm1.70}$ | $2.22_{\pm0.46}$ | $0_{\pm0}$ | $65.11_{\pm1.43}$ | $2.23_{\pm0.45}$ | $0_{\pm0}$ | $65.19_{\pm1.52}$ | $2.15_{\pm0.08}$ | $0_{\pm0}$ |
| TAGT | $64.82_{\pm0.61}$ | $2.35_{\pm0.37}$ | $0_{\pm0}$ | $63.82_{\pm0.76}$ | $2.10_{\pm0.34}$ | $0_{\pm0}$ | $64.31_{\pm0.97}$ | $2.18_{\pm0.28}$ | $0_{\pm0}$ |
| +ours | $66.13_{\pm1.17}$ | $2.55_{\pm0.34}$ | $1.95_{\pm1.15}$ | $66.20_{\pm1.08}$ | $2.58_{\pm0.29}$ | $1.78_{\pm1.82}$ | $66.35_{\pm1.17}$ | $2.57_{\pm0.21}$ | $2.05_{\pm2.08}$ |
| GraphMixer | $68.45_{\pm0.76}$ | $2.38_{\pm0.14}$ | $0_{\pm0}$ | $68.71_{\pm0.81}$ | $2.16_{\pm0.34}$ | $0_{\pm0}$ | $67.99_{\pm1.35}$ | $2.30_{\pm0.71}$ | $0_{\pm0}$ |
| CAWN | $60.67_{\pm2.33}$ | $1.53_{\pm0.19}$ | $0_{\pm0}$ | $66.15_{\pm2.01}$ | $2.02_{\pm0.50}$ | $0_{\pm0}$ | $66.34_{\pm0.50}$ | $1.68_{\pm0.07}$ | $0_{\pm0}$ |
| DyGFormer | $53.12_{\pm7.20}$ | $1.17_{\pm0.16}$ | $0_{\pm0}$ | $65.92_{\pm1.05}$ | $2.39_{\pm0.11}$ | $0_{\pm0}$ | $60.54_{\pm2.67}$ | $1.60_{\pm0.13}$ | $0_{\pm0}$ |
| FreeDyG | $62.30_{\pm2.15}$ | $1.44_{\pm0.31}$ | $0_{\pm0}$ | $67.11_{\pm0.97}$ | $2.13_{\pm0.21}$ | $0_{\pm0}$ | $61.54_{\pm1.30}$ | $2.06_{\pm0.53}$ | $0_{\pm0}$ |
| TCL | $63.01_{\pm1.31}$ | $1.95_{\pm0.12}$ | $0_{\pm0}$ | $64.95_{\pm0.47}$ | $2.49_{\pm0.37}$ | $0_{\pm0}$ | $65.67_{\pm0.93}$ | $1.99_{\pm0.27}$ | $0_{\pm0}$ |
| +ours | $64.38_{\pm1.08}$ | $2.04_{\pm0.13}$ | $1.63_{\pm0.26}$ | $66.02_{\pm0.39}$ | $2.70_{\pm0.22}$ | $1.66_{\pm1.41}$ | $67.17_{\pm0.63}$ | $2.79_{\pm0.81}$ | $1.62_{\pm1.17}$ |

*Table 13.* Performance comparison ($Avg_{\pm std}$ in %) on MOOC dataset under **S2**.

| L | Wikipedia | | | MOOC | | | $\alpha$ | Wikipedia | | | MOOC | | |
|---|---|---|---|---|---|---|---|---|---|---|---|---|---|
| | AUROC | AP | F1 | AUROC | AP | F1 | | AUROC | AP | F1 | AUROC | AP | F1 |
| 2 | $80.36_{\pm0.69}$ | **$5.41_{\pm1.23}$** | **$8.34_{\pm2.72}$** | $65.75_{\pm0.72}$ | $3.26_{\pm0.19}$ | **$8.74_{\pm0.54}$** | 0.001 | **$81.12_{\pm1.24}$** | $5.68_{\pm1.61}$ | $5.40_{\pm3.10}$ | $71.28_{\pm1.42}$ | $6.93_{\pm0.95}$ | **$7.42_{\pm2.81}$** |
| 5 | $83.76_{\pm0.49}$ | $5.36_{\pm0.53}$ | $5.33_{\pm3.23}$ | $68.51_{\pm1.86}$ | $3.33_{\pm0.16}$ | $4.27_{\pm1.83}$ | 0.005 | $80.77_{\pm0.84}$ | **$5.68_{\pm1.09}$** | $4.68_{\pm1.83}$ | $71.88_{\pm0.96}$ | $7.40_{\pm0.27}$ | $6.65_{\pm0.66}$ |
| 10 | $85.94_{\pm0.74}$ | $4.72_{\pm1.33}$ | $4.43_{\pm3.44}$ | $70.31_{\pm0.29}$ | $6.46_{\pm0.56}$ | $7.46_{\pm0.79}$ | 0.01 | $80.36_{\pm0.69}$ | $\underline{5.41_{\pm1.23}}$ | $\underline{8.34_{\pm2.72}}$ | $72.87_{\pm1.40}$ | **$7.89_{\pm0.38}$** | $7.15_{\pm0.64}$ |
| 20 | $87.22_{\pm0.76}$ | $4.94_{\pm0.85}$ | $4.67_{\pm4.04}$ | $72.87_{\pm1.40}$ | **$7.89_{\pm0.38}$** | $7.15_{\pm0.64}$ | 0.05 | $79.87_{\pm0.42}$ | $5.39_{\pm1.09}$ | $6.37_{\pm3.91}$ | **$72.91_{\pm0.45}$** | $7.12_{\pm0.40}$ | $5.43_{\pm0.43}$ |
| 32 | **$88.06_{\pm0.38}$** | $5.04_{\pm1.55}$ | $2.92_{\pm1.59}$ | **$73.56_{\pm1.11}$** | $7.24_{\pm0.44}$ | $6.20_{\pm0.38}$ | 0.1 | $79.70_{\pm0.89}$ | $5.37_{\pm0.91}$ | $6.46_{\pm2.68}$ | $72.73_{\pm0.81}$ | $7.00_{\pm1.03}$ | $5.78_{\pm1.04}$ |

| *(a)* Performance with different values of $L$. | *(b)* Performance with different values of $\alpha$. |

*Table 14.* Effect of hyperparameters $L$ and $\alpha$ on performance on Wikipedia and MOOC datasets. The best results are highlighted in **green** and the *underlined* results correspond to those reported in Table. 1.

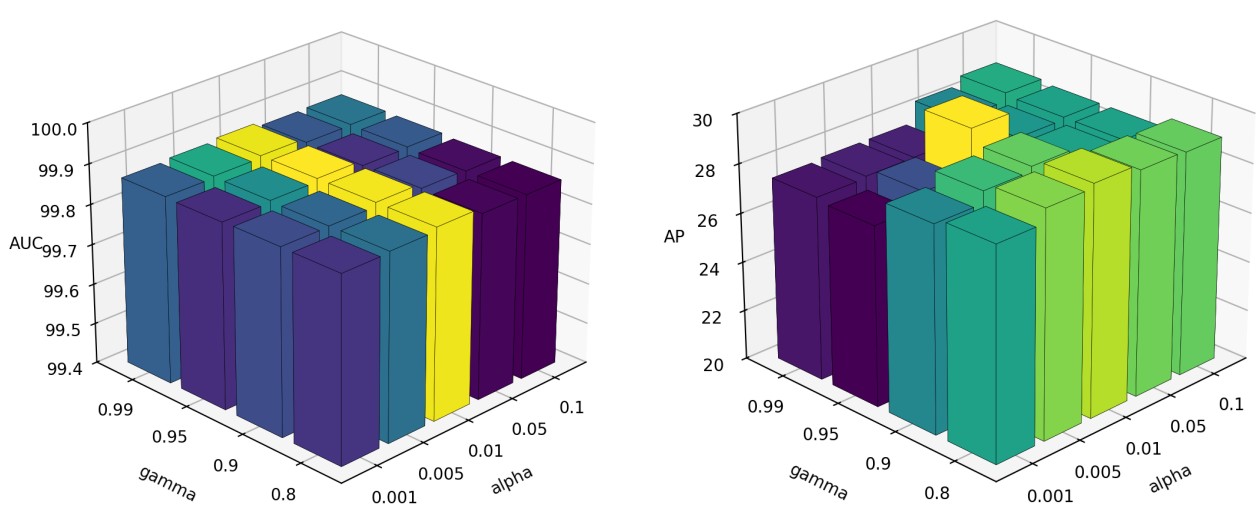

*Figure 6.* Parameter sensitivity with different $\alpha$ and $\gamma$ on Enron dataset

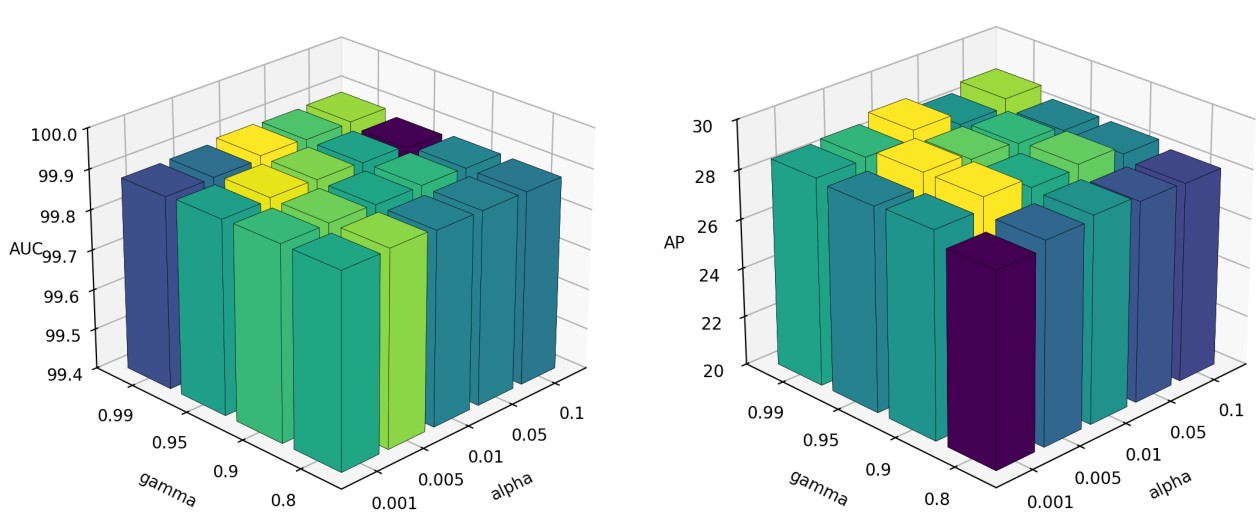

*Figure 7.* Parameter sensitivity with different $\alpha$ and $\gamma$ on UCI dataset

