# OpenReview forum: "Learning Discriminative and Generalizable Anomaly Detector for Dynamic Graph with Limited Supervision"
_ICML.cc/2026/Conference — ICML 2026 regular_

### Official Review · Reviewer_iFs4 · 2026-02-25

**Soundness:** 4
**Presentation:** 4
**Significance:** 3
**Originality:** 3
**Overall Recommendation:** 5
**Confidence:** 5

**Summary:**

This paper proposes SDGAD, a separation-driven framework for continuous-time dynamic graph anomaly detection. It enhances anomaly separability in both representation and likelihood spaces through three components: residual encoding for anomaly-sensitive representations, a norm restriction loss (LRR) to compact normal embeddings, and a bi-boundary likelihood optimization (LBO) built on normalizing flows to stabilize density-based separation. The framework is model-agnostic and evaluated under varying supervision regimes.

**Compliance With Llm Reviewing Policy:**

Affirmed.

**Final Justification:**

After reviewing the authors’ rebuttal and responses to other reviewers, I find that the clarifications adequately address my concerns. My overall assessment remains positive, and I therefore maintain my original positive score.

**Key Questions For Authors:**

please refer to weakness

**Limitations:**

yes

**Strengths And Weaknesses:**

S1: The paper clearly identifies that separability issues may arise both in representation learning and in likelihood estimation. The combination of residual encoding with norm restriction (LRR) and bi-boundary likelihood optimization (LBO) directly addresses this concern in a principled manner.

S2: SDGAD can be integrated with different dynamic graph encoders (e.g., TCL, CAWN, DyGFormer), demonstrating architectural flexibility and broad applicability.

S3: The experiments cover multiple datasets, including both real and synthetic anomaly settings, and evaluate performance under varying levels of supervision (S1–S3) as well as few-shot anomaly labels. The empirical study is thorough and systematically organized.

W1: Although the paper provides ablations for LRR and LBO, the contribution of the residual encoding module (Section 4.1) is not independently evaluated. Since the manuscript claims that residual encoding learns anomaly-sensitive representations, it would be important to verify whether this component itself significantly improves separability compared to standard representations. Without isolating this effect, it remains unclear how much of the performance gain stems from residual encoding versus the subsequent regularization and boundary optimization mechanisms.

W2: The claim of improving separability under “diverse anomaly patterns” in conclusion is not fully substantiated. While the method demonstrates robustness across datasets, the manuscript does not explicitly characterize or analyze distinct anomaly mechanisms. Therefore, the notion of diversity appears empirical rather than structurally grounded.

W3: The performance sensitivity to the LBO weight suggests that boundary enforcement may require dataset-specific tuning. Additional guidance on hyperparameter selection would improve practical robustness.

---

> ### Author Rebuttal · Authors · 2026-03-31
>
> Dear reviewer iFs4. Thanks for your overall positive assessment of our work. Below, we clarify the corresponding concerns in detail.
>
> **Weakness1: Ablation study of residual encoding module.**
>
> Thanks for the comment. We would like to clarify that the contribution of the residual encoding module is already evaluated independently in the current manuscript.
>
> Specifically, in Sec.5.3 / Table.3, the variant **w/o Res** removes the residual representation encoding while keeping the other components unchanged. As shown in Table.3, **w/o Res** leads to the largest overall degradation among all ablations, with consistent drops across AUROC, AP, and $F1$, as well as noticeably larger standard deviations. This result shows that residual encoding is an essential part of the framework and contributes not only to performance but also to training stability.
>
> We also provide a visualization in Sec.5.4 / Fig.4. The top-left subfigure shows the log-likelihood distribution produced by the baseline TCL with normalizing flow, where normal and anomalous samples still exhibit substantial overlap. The top-right subfigure further introduces residual encoding on top of this baseline, while the restriction loss and bi-boundary optimization are not yet applied. Compared with the top-left subfigure, the overlap is already reduced, indicating that residual encoding alone can improve separability before the later modules are introduced.
>
> **Weakness2: Claim in conclusion is not fully substantiated.**
>
> We thank the reviewer for this helpful comment. We agree that this claim is currently broader than our evidence supports. In our experiments, support for handling diverse anomaly patterns comes only from the synthetic datasets, where unseen anomaly types appear at test time (Appendix.B.2). On the real-world datasets, only binary anomaly labels are available, so this property cannot be strictly verified. We will therefore revise the conclusion to a more precise statement: “improve separability and support generalization to unseen anomaly types.”
>
> **Weakness3: The loss design is complex, the framework requires careful tuning of multiple hyperparameters. Additional guidance on hyperparameter selection would improve practical robustness.**
>
> We thank the reviewers for raising this common concern. Although our framework involves several hyperparameters, they are not independent arbitrary knobs. Instead, each parameter has a clear role, so tuning is guided by its effect on model behavior rather than by blind search.
>
> Specifically, $\lambda_1$ and $\lambda_2$ control the relative strengths of the different training objectives, and thus determine the balance among them. Smaller values weaken the corresponding objective, whereas overly large values may overemphasize one part of the training signal and reduce the overall balance. In addition, $r_{\max}$ is fixed at $0.4$ in our paper, and $r_{\min}$ is determined by $\gamma$ through $r_{\min} = \gamma r_{\max}$. Thus, $\gamma$ controls the strength of restriction on normal representations: a smaller $\gamma$ relaxes the restriction space and may weaken discriminability, whereas a larger $\gamma$ imposes tighter restriction. Likewise, $\alpha$ controls how conservative the normal boundary is, while $\tau$ controls the separation margin between normal and anomalous regions. Therefore, these hyperparameters govern a small number of interpretable aspects of the model, rather than introducing many unrelated degrees of freedom.
>
> More importantly, our results in Appendix.D and Appendix.E show that, in most cases, a single set of values already yields strong performance across different datasets. For example, $\tau = 0.1$ and $\alpha = 0.01$ work well across datasets, and performance remains stable when $\lambda_1$, $\lambda_2$, $\tau$, $\gamma$, $\alpha$, and $L$ vary within a reasonable range. This is even clearer on the synthetic datasets, where performance changes very little under different hyperparameter settings.
>
> At the same time, achieving the best possible performance on a particular dataset may still require further adjustment based on its anomaly characteristics. In essence, the key parameters reflect a trade-off between discriminability and generalization. If a dataset contains highly diverse anomaly patterns, especially when some anomalies are close to normal samples, overly strong restriction or an overly large separation margin may force the model to adopt an overly narrow notion of normality, which can hurt generalization to subtle anomalies. In such cases, smaller values of $\gamma$ or $\tau$ can provide more flexibility and improve generalization, although this may slightly reduce discriminability. By contrast, when anomalous patterns are more consistent and more clearly separated from normal ones, larger values of $\gamma$ or $\tau$ can improve separation.

---

> > ### Author Rebuttal · Reviewer_iFs4 · 2026-04-01
> >
> > Thank you for the clarification, and I will maintain my original positive score.

---

> > > ### Author Response · Authors · 2026-04-01
> > >
> > > Thank you very much for your positive feedback. We sincerely appreciate your time and support of our work.

---

### Official Review · Reviewer_qRys · 2026-03-12

**Soundness:** 4
**Presentation:** 3
**Significance:** 2
**Originality:** 2
**Overall Recommendation:** 4
**Confidence:** 3

**Summary:**

This paper proposes SDGAD, a supervised framework for dynamic graph anomaly detection that aims to improve discriminability under unsupervised-style challenges and generalization to unseen anomalies. The first component, Residual Representation Encoding, captures deviations from historical patterns by computing the difference between node embeddings with and without the current interaction. The second component, Representation Restriction, constrains the residuals of normal samples within an interval defined by two concentric hyperspheres, thereby promoting compact normal representations while preserving separability from anomalies. The third component, Bi-Boundary Optimization with Normalizing Flows, models the likelihood distribution of normal samples and introduces explicit normal and anomaly boundaries with a margin to enhance robustness.

**Compliance With Llm Reviewing Policy:**

Affirmed.

**Final Justification:**

The response has addressed my major concerns, and I have therefore raised my score to accept. I would also like to suggest that the authors add more discussion on the use of residual modules. Residual features have been widely explored in static graph anomaly detection methods such as ARC, AnomalyGFM, etc. The authors should clarify how their design differs from these existing approaches and further highlight the specific advantages of using such modules in the dynamic graph setting.

**Key Questions For Authors:**

(1) Why do the existing methods yield such low AP values across datasets? The authors should provide a detailed analysis of this phenomenon. It is particularly surprising that the F1 scores of most methods are 0 on several datasets.

(2) I am concerned about the training setting described in the paper. Specifically, $L_{ML}$ is defined as the maximum-likelihood objective of the normalizing flow and is computed only on normal samples to model the normal distribution, while $L_{BO}$ is constructed using both normal and abnormal batches. However, the paper claims to address an unsupervised setting without any labeled samples, as described in Section 5.3. If the method is applied in a fully unsupervised scenario, how are the normal and abnormal batches constructed for optimizing $L_{BO}$ and $L_{ML}$?

I will reconsider the overall score if the author clarifies the above questions.

**Limitations:**

yes

**Strengths And Weaknesses:**

**Strengths**

(1) The proposed method is effective on both real-world labeled anomalies (Wikipedia, Reddit, and
MOOC) and three benign datasets with injected anomalies

(2) The paper effectively articulates the limitations of unsupervised and semi-supervised approaches in DGAD, particularly the issue of ambiguous decision boundaries.

**Weaknesses**

(1) The proposed components are relatively standard and appear to be a straightforward combination of existing techniques, including residual representations, boundary-based loss, and normalizing flow.

(2) The loss design is fairly complex, as it combines residual regularization with bi-boundary optimization, which increases the tuning burden and introduces a large number of hyperparameters, even though the authors provide a sensitivity analysis. The framework requires careful tuning of multiple hyperparameters, including $\lambda_1$, $\lambda_2$, $r_{\min}$, $r_{\max}$, $\alpha$, and $\tau$, across different components, which raises concerns about the robustness and practical applicability of the method.

(3) The proposed method incorporates normalizing flow and bi-boundary optimization, both of which appear to be computationally expensive. This raises concerns about the scalability of the framework. The authors are encouraged to provide a complexity analysis, along with runtime comparisons against existing methods.

---

> ### Author Rebuttal · Authors · 2026-03-31
>
> Dear reviewer qRys. Thanks for your overall positive assessment of our work. Below, we clarify the corresponding concerns in detail.
>
> **Weakness1:The proposed components are relatively standard and appear to be a straightforward combination of existing techniques**
>
> Thanks for the comment.  Our contribution does not lie in claiming novelty for these components in isolation, but in their task-specific formulation and integration for dynamic graph anomaly detection task. Specifically, we show how residual representations can be used to capture anomalous deviations in dynamic graphs, how two decision boundaries can be derived from the learned likelihood distribution, and how these parts can be jointly optimized within a unified framework. These design choices are tailored to the characteristics of dynamic graph anomaly detection and do not follow from a simple combination of existing techniques.
>
>
> **Weakness2: The loss design is fairly complex,  the framework requires careful tuning of multiple hyperparameters:**
>
> Due to the character limit, please refer to our response to Weakness~3 raised by Reviewer iFs4 (the fourth reviewer).
>
> **Weakness3: The authors are encouraged to provide a complexity analysis, along with runtime comparisons against existing methods.**
>
> Due to the character limit, please refer to our response to Weakness~1 raised by Reviewer Edss (the first reviewer).
>
> **Question1: Why do the existing methods yield such low AP values across datasets?**
>
> We thank the reviewer for this important question. In fact, we have already provided a detailed analysis of this phenomenon in Appendix.F, and we are glad to clarify it again here because it is the main motivation of our paper.
>
> As stated in the Introduction (lines 44--53), many existing DGAD methods tend to produce anomaly scores that collapse into a narrow low-valued range, with substantial overlap between normal and anomalous samples.  This behavior is further illustrated in Fig.3 and Fig.7. As a result, normal and anomalous samples are not well separated in score space, which makes the decision threshold inherently ambiguous and unstable. This issue is especially severe under the highly imbalanced setting of anomaly detection, where negative (normal) samples vastly outnumber positive (anomalous) ones.
>
>
> In this setting, different evaluation metrics have very different sensitivity to false positives. As discussed in the Evaluation Metrics subsection (lines 322--328) and Appendix.F, AUROC can remain relatively high even when the ranking quality is not sufficient for practical detection, because it is computed in ROC space, where $\mathrm{FPR}=FP/(FP+TN)$. When $TN$ is very large, even a noticeable number of false positives may correspond to only a small increase in FPR. By contrast, AP is computed in PR space, where Precision $=TP/(TP+FP)$ is directly affected by false positives. Therefore, once high-score regions contain many normal samples due to score overlap, AP drops much more sharply.
>
> The same issue also explains why $F1$ is often $0$. Unlike AUROC or AP, $F1$ is evaluated at a predefined decision threshold. For example, as shown for TCL in Fig.3, even with a threshold of $0.01$, no anomalies can be detected because most anomaly scores remain below this value. As a result, recall becomes $0$, leading directly to $F1=0$.
>
>
> Therefore, the low AP and $F1$ values of existing methods mainly reflect insufficient score separation between normal and anomalous samples, especially under severe class imbalance. This is exactly the issue our method is designed to address by improving both representation discriminability and boundary quality.
>
> **Question2: Concerns about the training setting described in the paper.**
>
>  Thanks for raising this question. We would like to clarify that our framework is trained in a standard \emph{mini-batch} manner, rather than with separately constructed normal and abnormal batches. In our experiments, the batch size is set to $200$, and as described in Sec.~4.3 (lines 260--265), we denote by $N$ and $M$ the numbers of normal and anomalous samples in the mini-batch, respectively, with $N+M=200$.  The loss $L_{ML}$ is computed only on the normal samples within each mini-batch.
>
> Under the limited supervised setting, many mini-batches have $M=0$ because labeled anomalies are scarce. In this case, $L_{BO}$ reduces to its first term. When a mini-batch contains labeled anomalies ($M>0$), the second term of $L_{BO}$ is activated accordingly.
>
> The unsupervised setting is a special case of the same training procedure, where $M=0$ for every mini-batch.
>
> **We would like to thank you once again for you feedback and suggestion. If you still have any concerns or questions, please let us know and we will be happy to address them!**

---

> > ### Author Rebuttal · Reviewer_qRys · 2026-04-03
> >
> > Thank you for the response. I agree with the authors’ point that the limited separation between normal and anomalous samples in the score space is a common issue in unsupervised methods. I acknowledge the contribution of this paper to dynamic graph anomaly detection, and I will consider raising my score if the author could consider several suggestions I provide below
> >
> > (1) What thresholds were used in the main comparison?  Are they consistent across datasets?
> >
> > (2) The authors are encouraged to add an additional section on threshold analysis related to the reported TPR and FPR. In particular, it would be helpful to provide F1 score, FPR, and FNR under different thresholds, together with a discussion of the observed phenomena.

---

> > > ### Author Response · Authors · 2026-04-03
> > >
> > > We thank the reviewer for recognizing the contribution of our work to dynamic graph anomaly detection.
> > >
> > > To avoid any misunderstanding, we first clarify that AUROC and AP are threshold-independent metrics, as they evaluate model performance over all possible thresholds by sorting all predicted scores and treating each unique score as a candidate threshold. Therefore, here we focus only on the relationship among F1, TPR, and FPR under different thresholds.
> > >
> > > Before addressing your first question, we first clarify the training protocol used in all experiments. All models are trained for up to 200 epochs with early stopping (patience = 10), and the checkpoint with the best validation performance is selected for testing. Validation performance is also evaluated using AUROC, AP, and F1. Since F1 depends on a threshold $\epsilon$, we fix $\epsilon = 0.5$ during validation and use the same threshold to report test-time F1 for **all experiments**, ensuring a consistent and fair evaluation.
> > >
> > > Following your second suggestion, we reran SDGAD (TCL) and TCL on the Wikipedia and UCI datasets. During validation, F1 was still computed with $\epsilon = 0.5$. On the test set, we report F1, TPR, and FPR at thresholds of 0.01, 0.1, 0.3, and 0.5. The results are shown below.
> > >
> > > On the relatively simple synthetic anomaly dataset UCI, TCL yields $F1 = 0$, $TPR = 0$, and $FPR = 0$ under all tested thresholds, meaning that it fails to detect any anomalous samples. In contrast, SDGAD consistently achieves $F1 = 0.4444$, $TPR = 1$, and $FPR = 0.002298$.  In this case, all scores produced by TCL are below 0.01, making threshold-based discrimination ineffective. By contrast, our method gets much better separability.
> > >
> > > On the more challenging real-world anomaly dataset Wikipedia, although the effect is less pronounced than on UCI, a similar pattern can still be observed. Although TCL yields a low FPR at all tested thresholds, and its FPR further decreases as the threshold increases, its TPR also drops steadily. As a result, its overall F1 score remains poor. This indicates that TCL is not sensitive to threshold changes and provides limited class separability. In contrast, our method responds much more clearly to threshold variation: as the threshold increases, both TPR and FPR change substantially. This behavior indicates that our method achieves much better separability between anomalous and normal samples. Although our method has a higher absolute FPR than TCL, the trade-off between TPR and FPR is consistently more favorable for our method.
> > >
> > > In addition, we would like to proactively clarify two possible points of confusion.
> > >
> > > 1.  Reviewer may be concerned that, under a very low threshold (e.g., $\epsilon = 0.01$), our method yields an FPR close to 1, which may appear undesirable. However, we would like to clarify that this is a natural consequence of the much wider score range produced by our method. Once the score distribution is better spread out, lowering the threshold naturally leads to more positive predictions and thus a higher FPR. In our view, this should not be interpreted as a weakness, but rather as evidence that the model provides a meaningful and usable operating range, instead of collapsing to near-all-negative predictions.
> > >
> > > 2. All of the above analysis is based on the model checkpoint selected using validation F1 computed with $\epsilon = 0.5$, and the selected checkpoint is then evaluated on the test set. We acknowledge that, if a different validation threshold were used for model selection, the exact relationships among test-time F1, TPR, and FPR under different thresholds might change. However, we believe that the key conclusion would remain the same: our method improves separability and provides a much more usable threshold range for distinguishing anomalous from normal samples.
> > >
> > > **We will include these discussions, together with the complexity analysis raised by the reviewer in the first round, in the revised manuscript. We would like to again sincerely thank you for your constructive and insightful feedback, which has significantly improved the quality of our paper.**
> > >
> > > **TCL on UCI**
> > >
> > > | Threshold | F1 | TPR | FPR |
> > > |---|---:|---:|---:|
> > > | 0.01 | 0 | 0 | 0 |
> > > | 0.1 | 0 | 0 | 0 |
> > > | 0.3 | 0 | 0 | 0 |
> > > | 0.5 | 0 | 0 | 0 |
> > >
> > > **SDGAD on UCI**
> > >
> > > | Threshold | F1 | TPR | FPR |
> > > |---|---:|---:|---:|
> > > | 0.01 | 0.444444 | 1 | 0.002298 |
> > > | 0.1 | 0.444444 | 1 | 0.002298 |
> > > | 0.3 | 0.444444 | 1 | 0.002298 |
> > > | 0.5 | 0.444444 | 1 | 0.002298 |
> > >
> > > **TCL on Wikipedia**
> > >
> > > | Threshold | F1 | TPR | FPR |
> > > |---|---:|---:|---:|
> > > | 0.01 | 0.029557 | 0.073171 | 0.000555 |
> > > | 0.1 | 0.032520 | 0.024390 | 0.000620 |
> > > | 0.3 | 0.020408 | 0.012195 | 0.000238 |
> > > | 0.5 | 0.021277 | 0.012195 | 0.000175 |
> > >
> > > **SDGAD on Wikipedia**
> > >
> > > | Threshold | F1 | TPR | FPR |
> > > |---|---:|---:|---:|
> > > | 0.01 | 0.002664 | 1.0 | 0.976140 |
> > > | 0.1 | 0.006867 | 0.695122 | 0.261700 |
> > > | 0.3 | 0.031718 | 0.439024 | 0.034209 |
> > > | 0.5 | 0.111111 | 0.097561 | 0.000858 |

---

### Official Review · Reviewer_vXgc · 2026-03-12

**Soundness:** 2
**Presentation:** 2
**Significance:** 1
**Originality:** 1
**Overall Recommendation:** 1
**Confidence:** 5

**Summary:**

SDGAD proposes residual encoding, a bi-hypersphere restriction loss, and normalizing flow-based bi-boundary optimization for dynamic graph anomaly detection.

**Compliance With Llm Reviewing Policy:**

Affirmed.

**Final Justification:**

This paper claims to "propose the two co-centered hyperspheres" as a novel contribution for tightening support of the normal region. However, this construction clearly originates from existing work [1]. Authors also acknowledged this point in their responses. The theoretical foundation, descriptive framing, architecture (Fig. 2, bi-hypersphere part), and associated proofs bear substantial similarity to [1]. While the authors are expected to articulate their differentiation from [1], the rebuttal fails to do so. The only difference point I see is the application field is not the same.
The most important point is, it does not cite [1] appropriately. The only reference to [1] appears in a peripheral remark: "normal representations tend to concentrate near the sphere boundary instead of spreading throughout the interior, leaving the inner region weakly supported (the soap-bubble phenomenon [1])." This reference is entirely absent from the formal methodology, the architectural description, and any accompanying discussion, which misrepresents the provenance of a core technical contribution.
I consider this a significant violation of academic attribution norms. I maintain my current score.

**References:**

[1] "Deep Orthogonal Hypersphere Compression for Anomaly Detection." The Twelfth International Conference on Learning Representations 2024.

**Key Questions For Authors:**

Please refer to Weaknesses.

**Limitations:**

yes.

**Strengths And Weaknesses:**

**Strengths:**

1)	The paper is easy to follow.
2)	The experiments are well written.

**Weaknesses:**

1) Authors claim that “Therefore, we propose two co-centered hyperspheres to provide tighter support for the normal region.” However, the bi-hypersphere component, such as soap-bubble phenomenon, typical set concept, and the co-centered hypersphere construction, appears to be entirely adopted from [1]. The citation is insufficiently prominent, and the paper never explicitly delineates what is borrowed versus what is novel. The authors should provide a dedicated discussion of how their design differs from [1] and clearly justify their claimed contribution.

**References:**

- [1] "Deep Orthogonal Hypersphere Compression for Anomaly Detection." The Twelfth International Conference on Learning Representations 2024.
2) The motivation for choosing normalizing flows as the density estimator is not justified.
3) Parameter sensitivity of $r_{max}$ should be supplemented.
4) For anomalous samples, the loss penalizes representations inside radius $r_{max}$, but applies no penalty once they exceed $r_{max}$. This makes the anomaly loss unbounded from above on the normal side, raising a concern about convergence. What criterion indicates that the model has converged, and how is training stability ensured?
5) Normalizing flows require computing the Jacobian determinant at each forward pass, which introduces significant computational overhead. The paper provides no runtime or complexity comparison with baselines.
6) In Section B.2, only T-type anomalies are injected during training and validation, while the test set additionally contains S-type anomalies that are entirely unseen during training. Nevertheless, SDGAD achieves near-perfect AUROC under the fully unsupervised setting. It is unclear how the model achieves perfect separation on anomaly types it has never encountered, especially without any anomaly supervision.
7) A key ablation is missing: using the normalizing flow log-likelihood alone as the anomaly score, without $\mathcal{L}\_{RR}$ or $\mathcal{L}\_{BO}$.

---

> ### Author Rebuttal · Authors · 2026-03-31
>
> Dear reviewer vXgc. We are pleased that you find our paper is easy to follow and that the experimental section is clearly presented. Below, we address the concerns raised in the weakness section and clarify the corresponding points in detail.
>
> **W1:** Thanks for your comment. However, we respectfully disagree that our design is ``entirely adopted'' from [1]. We cite [1] to acknowledge the geometric motivation that, in high-dimensional space, the soap-bubble phenomenon may make a single hypersphere provide weak support for the normal region, so an annular region can be more suitable. However, sharing this intuition does not mean that the two methods are the same.
>
> Our method differs from [1] in three key aspects.
>
> First, the construction of the two hyperspheres is different. In [1], they are obtained through a data-driven multi-stage optimization procedure: it first learns a representation space and a center, derives a single-hypersphere solution, and then further optimizes to obtain the second hypersphere. As a result, the learned hyperspheres depend strongly on the training data and require a relatively cumbersome optimization process. In contrast, we directly define the outer hypersphere as the unit hypersphere under the Euclidean norm and the inner hypersphere with a single hyperparameter, making our construction much simpler and easier to use.
>
> Second, the loss design is different. In [1], the bi-hypersphere is optimized with a hard penalty under a purely unsupervised objective. By contrast, our method uses soft penalties and further incorporates an anomaly-supervised term when anomaly labels are available.
>
> Third, the role of the bi-hypersphere module is different. In [1], the learned two-hypersphere region is directly used for final scoring and decision making. In our framework, the co-centered hyperspheres are used only to restrict normal residual representations for improving separability, and reducing overfitting. The final anomaly decision is made later in the likelihood-space boundary learning stage.
>
> We thank the reviewer for this suggestion and will revise the paper to clarify the connection to [1], make the attribution more explicit, and state our contribution more clearly.
>
> **W2**: As stated in Sec.4.3, our goal is to model the normal distribution and then learn a decision boundary in log-likelihood space. Normalizing flows are a natural choice because they provide exact density estimation and explicit log-likelihoods for boundary learning.
>
> **W3**:  We would like to clarify that $r_{\max}$ is not a tunable hyperparameter. As stated in Sec.4.2 (lines 183--186), we measure the residual norm by the pseudo-Huber surrogate $n(x)$, and the outer hypersphere is defined to align with the unit hypersphere under the Euclidean norm. So we fix $r_{\max} =0.4$,since $n(x)\approx 0.4$ when ${\|x\|_2^2=1}$.
>
> **W4**: Thanks for your question. The anomaly loss being unbounded from above does not itself cause a convergence issue, since training minimizes the objective. What matters is that it is lower-bounded. In our case, the anomaly term is non-negative and becomes 0 once an anomalous representation lies outside the target radius, so it does not prevent convergence. Moreover, this term is active only when anomalous samples are mapped inside the normal region, where it pushes them outward. Once they cross the radius, the penalty vanishes. In practice, convergence is determined by the stabilization of the overall objective and the validation criterion.
>
> **W5**: Due to the character limit, please refer to our response to Weakness~1 raised by Reviewer Edss (the first reviewer).
>
> **W6**: Thanks for your question. We would first like to clarify that a near-perfect AUROC does not imply perfect absolute separation; it only indicates strong ranking ability of the anomaly scores. We refer the reviewer to Appendix.F for a more detailed analysis of the evaluation metrics.
>
> More importantly, generalization to unseen anomaly types is exactly the goal of our method. Under the fully unsupervised setting, SDGAD is not trained to recognize specific anomaly types, but to model normality and assign higher scores to samples that deviate from it.  In addition, as discussed in Appendix.A, the synthetic anomaly types used in current benchmarks are relatively simple, which may make even unseen anomaly types easier to distinguish once they deviate sufficiently from the learned normal structure.
>
> **W7**: We would like to clarify that this ablation is already included in Sec.5.4 / Fig.4. Specifically, the top-left subfigure in Fig.4 corresponds to the TCL + normalizing flow baseline. The remaining three subfigures then progressively add residual encoding, the restriction loss, and bi-boundary optimization.
>
> **We would like to thank you once again for you feedback. If you still have any concerns or questions, please let us know and we will be happy to address them. Otherwise, we will appreciate it if you would reconsider your score.**

---

> > ### Author Rebuttal · Reviewer_vXgc · 2026-04-04
> >
> > Thank you for the detailed rebuttal. After carefully review, some of my concerns still exist:
> >
> > For W1, the three distinctions offered in the rebuttal are real at the implementation level, but they do not address the substance of my original concern. My concern was not about downstream usage or loss formulation details, but about whether the theoretical motivation, geometric intuition, and structural design of the bi-hypersphere component constitute an original contribution of this paper.
> >
> > Section 4.2 of the submission and Section 3.1 of [1] present nearly identical conceptual arguments: both cite Vershynin (2018), both invoke the soap-bubble phenomenon by name, both use the term "co-centered hyperspheres," and both argue that the annular region between two hyperspheres provides tighter support for the normal data distribution. This shared conceptual core is precisely what my original concern identified.
> > Regarding the three specific points raised in the rebuttal:
> >
> > On construction: The claimed difference "SDGAD fixes the outer radius analytically while [1] derives it from data"  is an implementation detail rather than a conceptual distinction. The core idea of confining normal representations to a hyperspherical shell, motivated by the soap-bubble phenomenon, originates in [1].
> >
> > On loss design: The use of soft rather than hard penalties is a minor engineering choice. That SDGAD adds an anomaly-supervised term is a genuine difference, but this pertains to the semi-supervised extension, not to the bi-hypersphere construction itself.
> >
> > On functional role: This is the most substantive distinction raised, i.e., using the bi-hypersphere as an intermediate regularizer rather than a final scoring boundary is a meaningful architectural difference. However, this does not resolve the attribution concern. A component can serve a different downstream role while still being conceptually borrowed from prior work. The paper should explicitly acknowledge what is inherited from [1] and clearly articulate why the changed functional role within the SDGAD framework constitutes a novel contribution beyond straightforward application of an existing idea.
> >
> > For W2, the rebuttal restates what is already in the paper, namely that normalizing flows provide exact density estimation and explicit log-likelihoods, without explaining why they are preferable over alternative density estimators such as GMMs, KDE, or energy-based models in this specific setting. No comparative justification or supporting ablation is provided.y estimators (GMM, KDE, energy-based models) in this specific setting. No comparative justification or ablation is provided.
> >
> > For W6, the authors suggest that near-perfect AUROC on unseen S-type anomalies may be attributable to the relative simplicity of synthetic anomaly types in current benchmarks. This argument effectively questions the benchmark's difficulty rather than explaining the model's generalization mechanism, and therefore undermines rather than strengthens the experimental claims. The principled reason why a model trained exclusively on T-type anomalies generalizes so effectively to unseen S-type anomalies remains unexplained.
> >
> > In summary, a proper response to my concern would need to either explicitly delineate what is adopted from [1] versus what is novel, or argue more rigorously that the integration of the bi-hypersphere component into the normalizing flow framework constitutes a conceptually new contribution. The current rebuttal does neither. I therefore maintain my original score.

---

> > > ### Author Response · Authors · 2026-04-08
> > >
> > > Thanks for your feedback. We now provide further clarification on the remaining concerns below.
> > >
> > > For W1, we would like to further clarify this point from three perspectives.
> > >
> > > **Hypersphere construction:** [1] is built on the assumption that samples are independent and identically distributed (i.i.d.). Under this assumption, [1] first learns a representation space and a center from carefully selected training samples, and then learns the two hyperspheres in separate optimization stages. As a result, its construction is computationally expensive, not end-to-end, and its effectiveness depends on the test data satisfying the same i.i.d. assumption. In contrast, samples in dynamic graphs are generally non-i.i.d., **Therefore, developing an effective and computationally efficient bi-hypersphere construction that can be learned end-to-end for dynamic graph anomaly detection is a new technical challenge rather than a direct extension of [1], and our solution constitutes a distinct contribution**.
> > >
> > > **Functional role:** The bi-hypersphere mainly imposes a geometric constraint by assessing whether a sample lies in the expected region of the representation space, whereas the flow-based likelihood captures distributional similarity by evaluating whether the sample is consistent with the learned distribution of normal data. As a result, a sample may lie within the annular region defined by the bi-hypersphere yet still receive a low likelihood, because normal samples may concentrate in only a few subregions rather than being uniformly distributed across the entire region. Similarly, a sample may receive a high likelihood because its local feature pattern remains consistent with the learned normal distribution, while its representation still falls outside the expected region. Therefore, **integrating the bi-hypersphere component into the normalizing-flow framework enables the model to reject samples that satisfy only one of the two criteria, yielding a more expressive anomaly scoring criterion than either component alone**.
> > >
> > > **Conceptual contribution:** In the paper, we acknowledged the contribution of [1] and cited it as part of the motivation for our design. However, as discussed above, our bi-hypersphere construction and its functional role differ substantially from those in [1], and we therefore do not regard the two methods as the same. **Nevertheless, we understand the reviewer’s concern that the underlying bi-hypersphere conceptual contribution was introduced in [1]. We will revise Section 4.2 to explicitly acknowledge this conceptual origin of bi-hypersphere and to more clearly distinguish our method from [1] in terms of both hypersphere construction and functional role.**
> > >
> > >
> > >
> > > For W2, our setting requires a density estimator that (i) provides explicit per-sample likelihoods for anomaly scoring, (ii) remains tractable during both training and inference, and (iii) is expressive enough to capture complex non-Gaussian feature distributions. While GMMs provide explicit likelihoods, their parametric form is relatively restrictive and may require many components to approximate strongly non-Gaussian structure. KDE becomes both statistically unreliable and computationally expensive in high-dimensional spaces. Energy-based models typically require approximation of the partition function and therefore do not yield explicit likelihoods. In this context, normalizing flows provide a practical balance between tractability, expressiveness, and explicit likelihood estimation.
> > >
> > >
> > > For W6, regarding the generalization mechanism, it is enabled by the representation restriction module together with the normalizing-flow-based boundary optimization. As discussed in the functional-role part of our response to W1, our method is designed to characterize normality from complementary geometric and distributional perspectives. As a result, unseen S-type anomalies can be detected if they deviate from the learned notion of normality. In the limited-supervision setting, the observed T-type anomalies primarily help refine the normality boundary, rather than making the model dependent on a particular anomaly type. This is the principled reason why the model generalizes beyond the anomaly types observed during training.
> > >
> > >
> > > Our comment on the relative simplicity of synthetic anomalies was meant as an empirical observation about the current **widely used anomaly synthesis protocol**, not as an explanation of the model’s generalization mechanism. As the reviewer noted, our method already achieves near-perfect AUROC performance under the widely used synthesis setting, suggesting that constructing more realistic and challenging synthetic anomalies is an important direction for future work. We also discussed this limitation in Appendix A. And this does not undermine our experimental claims, since we evaluate on both synthetic and real anomalies, and the method performs well in both settings.

---

### Official Review · Reviewer_Edss · 2026-03-13

**Soundness:** 3
**Presentation:** 3
**Significance:** 3
**Originality:** 3
**Overall Recommendation:** 4
**Confidence:** 2

**Summary:**

The paper presents a theoretically sound and well-motivated approach to dynamic graph anomaly detection. The residual encoding and bi-boundary optimization are elegant solutions to boundary ambiguity. Although I am not in this domain of research in time series anomaly detection representation restriction often hampers generalization capabilities. It is surprising that in this domain representation restriction is having good generalization. Can the authors kindly throw some light on this?

**Compliance With Llm Reviewing Policy:**

Affirmed.

**Final Justification:**

Rebuttal answered all concerns.

**Key Questions For Authors:**

How can you explain the role of restrictive representation in generalization?

**Limitations:**

Not discussed

**Strengths And Weaknesses:**

Strengths:

 The SDGAD framework is highly versatile and can be seamlessly integrated with various existing continuous-time dynamic graph (CTDG) base encoders.

Computing the residual representation explicitly captures the deviations between current interactions and historical contexts, isolating anomaly-relevant signals effectively.

 The use of a representation restriction loss bounded by two co-centered hyperspheres is a clever way to ensure consistent scales while avoiding the "soap-bubble phenomenon" seen in standard one-class classification.

 The bi-boundary optimization strategy, which models the normal log-likelihood distribution via a normalizing flow, effectively resolves the ambiguous scoring issues common in unsupervised methods.

 The paper presents extensive experiments across six datasets (featuring both real and synthetic anomalies) and demonstrates consistent superiority across diverse evaluation settings.


Weaknesses

 Utilizing Normalizing Flows for density estimation on continuous-time event streams introduces significant computational overhead. The paper lacks a deep discussion on inference latency for real-time deployment.

Because the framework is model-agnostic and relies on differencing features, its fundamental expressivity and performance ceilings are inherently bottlenecked by the capability of the chosen underlying CTDG encoder.

The framework introduces multiple sensitive hyperparameters, including the decision margin $\tau$, restriction strength $\gamma$, percentile $\alpha$, and history length $L$. Tuning these robustly is extremely difficult in real-world scenarios with extreme label scarcity.

---

> ### Author Rebuttal · Authors · 2026-03-31
>
> Dear reviewer Edss. Thanks for your overall positive assessment of our work. Below, we clarify the corresponding concerns in detail.
>
> **Weakness1: Complexity analysis and inference latency**
>
> Thanks for raising this concern.
>
> Let $B$ be the batch size, and let $T_{\\mathrm{enc}}(L)$ denote the cost of one CTDG encoder call on a node history of length $L$. For a baseline with the same backbone, each event requires encoding the two endpoints once, so the per-batch cost is
> $$
> T_{\\mathrm{base}} = O\\left(2B\\,T_{\\mathrm{enc}}(L)\\right).
> $$
> In SDGAD, the main additional cost comes from residual construction in Eq.~(3), where each endpoint is encoded twice. This increases the encoder-side cost to
> $$
> O\\left(4B\\,T_{\\mathrm{enc}}(L)\\right).
> $$
> The remaining overhead is comparatively small. The restriction module only adds a linear projection with cost $O(B\\,d_r d_x)$, where $d_r$ and $d_x$ denote the residual and projected dimensions. For the normalizing flow, we would like to clarify that the Jacobian determinant does not require forming a full dense Jacobian at each forward pass. Instead, we use coupling-based invertible blocks, for which the log-determinant can be computed analytically from the scaling terms. In our implementation, the flow consists of $6$ coupling blocks with two-layer fully connected subnetworks, resulting in a cost of $O(6B\\,d_x^2)$. Therefore, the total per-batch complexity is
> $$
> T_{\\mathrm{SDGAD}} = O\\left(4B\\,T_{\\mathrm{enc}}(L) + B\\,d_r d_x + 6B\\,d_x^2\\right).
> $$
> Thus, the dominant cost is still $T_{\\mathrm{enc}}(L)$. The additional costs from the restriction module and the flow depend only on the representation dimensions.
>
> Importantly, although the complexity expression suggests more than a $2\\times$ increase, the two branches in residual construction are independent and can be executed in parallel. As a result, the actual increase in inference time is much smaller than the worst-case complexity expression may suggest. We also provide empirical runtime on the MOOC dataset, comparing SDGAD (with TCL as the encoder) against TCL. The average inference time is $13 s$ for SDGAD versus $9s$ for TCL, which shows that the practical overhead is modest.
>
> **Weakness2: Performance ceilings are inherently bottlenecked.**
>
> Thanks for your comment. We agree that the underlying CTDG encoder affects the absolute performance ceiling.  However, this is orthogonal to our contribution: The encoder determines the basic quality of representations, while our framework builds on these representations to derive more anomaly-sensitive signals and learn a discriminative and robust boundary.
>
> **Weakness3: Multiple sensitive hyperparameters.**
>
> Due to the character limit, please refer to our response to Weakness~3 raised by Reviewer iFs4 (the fourth reviewer).
>
> **Question1: Research in time series anomaly detection representation restriction often hampers generalization capabilities.**
>
> Thank you for raising this interesting question. We would like to clarify this from two aspects.
>
> First, we think the key difference lies in how an individual sample is constructed and how anomaly is defined on it. In time series anomaly detection, an individual sample is usually constructed through preprocessing, such as sliding windows, segments, or subsequences, rather than taken directly from the original data format. Meanwhile, anomalies in time series may be defined at very different granularities, such as a single timestamp, a short anomalous interval or a contextual deviation. Consequently, the sample often does not align with the anomaly definition. A single sample may contain both normal and anomaly-related content, making its representation inherently mixed. Applying representation restriction to such mixed samples may distort anomaly-related information and reduce the separability, thereby making generalization more difficult. In contrast, in dynamic graph anomaly detection, each sample naturally corresponds to an interaction event between two nodes at a specific timestamp, without additional preprocessing. The task is then to determine whether the event is anomalous. In this sense, the sample and the anomaly definition are directly aligned.
>
> Second, we would like to clarify that we do not claim that representation restriction alone leads to strong generalization. In our framework, residual encoding extracts anomaly-relevant information, but different anomaly patterns can induce residuals with very different scales, which makes it difficult to learn a unified boundary directly.  This is precisely why restriction is helpful in our framework: it prepares a more regular normal space on top of which the bi-boundary objective can learn a more clear and robust separation. Therefore, the final generalization comes from the overall design of residual encoding, restriction, and bi-boundary optimization.
>
> **If you still have any concerns or questions, please let us know and we will be happy to address them!**

---

> > ### Author Rebuttal · Reviewer_Edss · 2026-04-01
> >
> > My concerns were addressed. I keep my initial positive score

---

> > > ### Author Response · Authors · 2026-04-01
> > >
> > > Thank you very much! We sincerely appreciate your time and support of our work！

---

### Decision · Program_Chairs · 2026-04-30

**Decision:**

Accept (regular)

**Comment:**

Based on the reviews, rebuttal, and discussion, I recommend weak acceptance. Reviewers iFs4, Edss, and qRys found the paper technically sound, relevant to dynamic graph anomaly detection, and empirically strong across multiple backbones, datasets, and supervision settings. The rebuttal resolved most technical concerns, including the residual-encoding ablation, the fully unsupervised training protocol, the threshold analysis, and the need to narrow the generalization claim.

The main remaining issue is originality and attribution of the co-centered hypersphere component. Reviewer vXgc raised a serious high-confidence concern that this idea is too close to prior work [1] and is not credited clearly enough. I find this concern important. However, I do not view it as fatal: even with narrower claims around this component, the paper still provides a useful dynamic-graph-specific integration of residual encoding, representation restriction, and likelihood-space boundary learning. A secondary concern is method complexity and tuning burden. The rebuttal partly addressed this, but the final version should present runtime and scalability tradeoffs more clearly.

If accepted, the final version should explicitly credit the conceptual origin of the bi-hypersphere idea in [1], clearly separate borrowed elements from new ones, and narrow the novelty claims accordingly.